# Comprehensive Evaluation of Ten *Actinidia arguta* Wines Based on Color, Organic Acids, Volatile Compounds, and Quantitative Descriptive Analysis

**DOI:** 10.3390/foods12183345

**Published:** 2023-09-06

**Authors:** Jinli Wen, Yue Wang, Weiyu Cao, Yanli He, Yining Sun, Pengqiang Yuan, Bowei Sun, Yiping Yan, Hongyan Qin, Shutian Fan, Wenpeng Lu

**Affiliations:** Institute of Special Animal and Plant Sciences, Chinese Academy of Agricultural Sciences, Changchun 130112, Chinaqinhongyan@caas.cn (H.Q.); fanshutian@caas.cn (S.F.)

**Keywords:** *Actinidia arguta* wine, color, organic acids, volatile flavor components, HS-GC-IMS, odor activity value, key aroma compounds, quantitative descriptive analysis

## Abstract

*Actinidia arguta* wine is a low-alcoholic beverage brewed from *A. arguta* with a unique flavor and sweet taste. In this study, the basic physicochemical indicators, color, organic acid, and volatile aroma components of wines made from the *A. arguta* varieties ‘Kuilv’, ‘Fenglv’, ‘Jialv’, ‘Wanlv’, ‘Xinlv’, ‘Pinglv’, ‘Lvbao’, ‘Cuiyu’, ‘Tianxinbao’, and ‘Longcheng No.2’ were determined, and a sensory evaluation was performed. The findings show that ‘Tianxinbao’ produced the driest extract (49.59 g/L), ‘Kuilv’ produced the most Vitamin C (913.46 mg/L) and total phenols (816.10 mg/L), ‘Jialv’ produced the most total flavonoids (477.12 mg/L), and ‘Cuiyu’ produced the most tannins (4.63 g/L). We analyzed the color of the *A. arguta* wines based on *CIEL***a***b** parameters and found that the ‘Kuilv’ and ‘Longcheng No.2’ wines had the largest *L** value (31.65), the ‘Pinglv’ wines had the greatest *a** value (2.88), and the ‘Kuilv’ wines had the largest *b** value (5.08) and C**_ab_* value (5.66) of the ten samples. A total of eight organic acids were tested in ten samples via high-performance liquid chromatography (HPLC), and we found that there were marked differences in the organic acid contents in different samples (*p* < 0.05). The main organic acids were citric acid, quinic acid, and malic acid. The aroma description of a wine is one of the keys to its quality. A total of 51 volatile compounds were identified and characterized in ten samples with headspace gas chromatography-ion mobility spectrometry, including 24 esters, 12 alcohols, 9 aldehydes, 3 aldehydes, 2 terpenes, and 1 acid, with the highest total volatile compound content in ‘Fenglv’. There were no significant differences in the types of volatile compounds, but there were significant differences in the contents (*p* < 0.05). An orthogonal partial least squares discriminant analysis (OPLS-DA) based on the odor activity value (OAV) showed that ethyl butanoate, ethyl pentanoate, ethyl crotonate, ethyl isobutyrate, butyl butanoate, 2-methylbutanal, ethyl isovalerate, and ethyl hexanoate were the main odorant markers responsible for flavor differences between all the *A. arguta* wines. Sensory evaluation is the most subjective and effective way for consumers to judge *A. arguta* wine quality. A quantitative descriptive analysis (QDA) of the aroma profiles of ten grapes revealed that the ‘fruity’ and ‘floral’ descriptors are the main and most essential parts of the overall flavor of *A. arguta* wines. ‘Tianxinbao’ had the highest total aroma score. The flavor and quality of *A. arguta* wines greatly depend on the type and quality of the *A. arguta* raw material. Therefore, high-quality raw materials can improve the quality of *A. arguta* wines. The results of the study provide a theoretical basis for improving the quality of *A. arguta* wines and demonstrate the application prospects of HS-GC-IMS in detecting *A. arguta* wine flavors.

## 1. Introduction

*Actinidia arguta* (mini kiwi) is a dioecious plant native to China predominantly found in the northeast, northwest, and north [1]. It is also distributed in Russia, Japan, Korea, the United States, New Zealand, and other regions [2,3]. *A. arguta* fruit is sweet, sour, fragrant, delicious and rich in a variety of vitamins, amino acids, proteins, minerals, and other nutrients [2], in addition to flavonoids, polysaccharides, volatile oils, and other biologically active ingredients [4,5]. *A. arguta* fruit has anti-oxidation and anti-tumor properties. Furthermore, it can lower blood sugar and enhance immunity, as well as having other health benefits [6,7,8,9,10]. Thanks to these properties, it has gained a reputation as a superfood [11] and is popular among consumers [12,13]. The Institute of Special Animal and Plant Sciences of the Chinese Academy of Agricultural Sciences has collected *A. arguta* germplasm resources and conducted variety selection and breeding since the 1960s. It has selected and bred high-quality varieties to promote the development of the *A. arguta* industry, such as ‘Kuilv’, ‘Fenglv’, ‘Jialv’, ‘Wanlv’, ‘Xinlv’, ‘Pinglv’, ‘Lvbao’, ‘Cuiyu’, ‘Tianxinbao’, and ‘Ruilv’ [14]. However, *A. arguta* is a climacteric respiration fruit with a thin skin so it is hard to store [11,15]; therefore, processing it into wine can consume non-commercial fruits and overproduction to alleviate spoilage problems and increase the added value of the product [16]. *A. arguta* wine is a new type of fruit wine. As raw materials, its fruits are brewed into a low-alcohol beverage, retaining *A. arguta*’s nutrient, anti-oxidant, anti-aging, and immunity-improving properties, as well as other effects [17,18]. The flavor and quality of *A. arguta* wine depend largely on the type and quality of the raw materials. Physicochemical indicators, color, organic acids, and volatile compounds are important factors that determine the quality of *A. arguta* wine [13].

The separation and quantitative determination of organic acids in food can be carried out using gas chromatography (GC), ion exchange chromatography (IEC), high-performance liquid chromatography (HPLC), and other methods [19]. Of these, HPLC is the most developed [20]. HPLC has the characteristics of high separation efficiency, fast analysis speed, and good detection sensitivity, as well as the ability to analyze and separate thermally unstable, physiologically active substances with high boiling points that cannot be vaporized. It has become an important separation and analysis technology in the fields of chemistry, medicine, industry, agronomy, forensic science, and other disciplines [20,21,22].

In recent decades, common detection techniques used in food aroma research have mainly involved instrumental detection (e.g., gas chromatography–mass spectrometry) and sensory detection (e.g., gas chromatography olfactometry), which have been used to analyze the content and intensity magnitude of aroma substances [23]. The detection of volatile substances in fruit wines is generally performed via gas chromatography–mass spectrometry (GC-MS), which has the advantages of being a mature technology and having a complete spectrum library, but the sample pretreatment is cumbersome and prone to error [24]. Gas chromatography–ion mobility spectrometry (GC-IMS) generates two-dimensional maps of volatile compounds based on the retention characteristics of the GC column and the ion mobility of the IMS detector, which is much more convenient, quicker, and more accurate. In addition, the sample does not require complex concentration and enrichment, which is conducive to maintaining the stability of flavor substances. Therefore, it can be widely used in differentiating volatile components and isomers, the analysis of trace components, and rapid on-site detection [25,26,27].

Although a series of works on *A. arguta* wine has been carried out by other researchers—such as on the effects of saccharomyces cerevisiae [28,29,30], process optimization [31,32], and varieties [33] on the quality of *A. arguta* wine—studies on the flavor of *A. arguta* wine need improvement. First, HS-GC-IMS has been widely used to determine volatile compounds in food [23,25,26,34]. However, the use of HS-GC-IMS to determine the volatile substances in *A. arguta* wines has not been reported. Second, previous research has only analyzed the volatile compounds in *A. arguta* wine and failed to identify the key volatile compounds [35]. Last, there has been a lack of sensory evaluation, which is how customers can best evaluate the quality of *A. arguta* wine. 

In this study, ten *A. arguta* varieties were harvested as the raw material for wine production in 2022 in Zuojia Town, Jilin City, Jilin Province, China. The basic physicochemical indicators, color indicators, and volatile compounds were determined, and sensory evaluation was performed. The fingerprints of volatile compounds in different varieties of *A. arguta* wine were also established. Moreover, the specific *A. arguta* wine aroma characteristics were characterized based on multivariate statistical analysis and quantitative descriptive analysis of the volatile compound data. This was combined with principal component analysis, OAV (odor activity value) analysis, and VIP (variable importance in projection) analysis to screen the key volatile compounds affecting *A. arguta* wine aroma and identify the volatile compounds that may affect wine flavor. The purpose of this study is to provide a theoretical basis for scientifically understanding the flavor and chemical nature of the aroma characteristics of different varieties of *A. arguta* to improve the quality of its wine and select winemaking fruit varieties.

## 2. Materials and Methods

### 2.1. Materials and Reagents

#### 2.1.1. Sample Preparation

Ten *A. arguta* varieties, including ‘Kuilv’, ‘Fenglv’, ‘Jialv’, ‘Wanlv’, ‘Xinlv’, ‘Pinglv’, ‘Lvbao’, ‘Cuiyu’, ‘Tianxinbao’, and ‘Longcheng No.2’, were used in this study. *A. arguta* fruits are shown in Appendix A. The sampling time was September 2022 when the fruits were ripe. The sampling site was the *Actinidia arguta* Resource Nursery of the Institute of Special Animal and Plant Sciences of the Chinese Academy of Agricultural Sciences, Zuojia Town, Jilin City, Jilin Province. 

The resource nursery is on gently sloping land in the mountains, with dark brown forest soil. After the soil thawed in the spring, a planting hole with a diameter of 40 cm and a depth of 30 cm was excavated in the middle of a planting ditch, where the ten *A. arguta* varieties were planted. The seedlings’ roots were evenly dispersed in the hole. We used a T-shaped frame for cultivation, row spacing of 3.5 m × 2.0 m, and a male and female plant configuration ratio of 8:1. We fertilized 2–3 times a year and weeded 3–4 times a year. Sampling was performed by randomly selecting well-grown, medium-sized fruit trees in the resource nursery, choosing soft date palm kiwifruit with the same degree of exposure to light, the same size, and similar hardness and fruit that was free of pests and diseases. We picked 20 kg of fruit from each variety, placed the samples in a sampling bag, and transported them back to the laboratory in an insulated box for winemaking.

#### 2.1.2. Reagents

Fermentation auxiliaries: BV818 active dry yeast (Angel Yeast Co., Ltd., Yichang, China); methanol fine-control enzyme RF (HaiDing Tang Guoji Trade Co., Ltd., Shanghai, China); and potassium metabisulfite (Shanghai Titan Scientific Co., Ltd., Shanghai, China).

Analytical purity reagents: sodium hydroxide, anhydrous sodium carbonate, anhydrous sodium acetate, ferrous sulfate, sulfuric acid, hydrochloric acid (Beijing Chemical Factory, Beijing, China); sodium acetate, anhydrous glucose (Xilong Chemical Co., Ltd., Guangzhou, China); gallic acid, tannic acid (Tianjin Guangfu Fine Chemical Research Institute, Tianjin, China); Folin–Denis (Sigma-Aldrich Co., Ltd., St. Louis, MO, USA); rutin (HaiDing Tang Guoji Trade Co., Ltd., Shanghai, China); Folin–Ciocalteu phenol reagent (Beijing Solarbio Science & Technology Co., Ltd., Beijing, China); anthrone, aluminum nitrate ninhydrate, potassium chloride, ferric chloride, pyrogallol, anhydrous ethanol, phenolphthalein (Sinopharm Chemical Reagents Co., Ltd., Shanghai, China).

Chromatographic purity standards: vitamin C (Shanghai Standard Technology Co., Ltd., Shanghai, China); oxalic acid, malic acid, shikimic acid, citric acid, succinic acid, quinic acid (Shanghai Yuanye Biotechnology Co., Ltd., Shanghai, China); methanol (TEDIA Reagents, Fairfield, OH, USA); 4-methyl-2-pentanol (Shanghai Aladdin Biochemical Technology Co., Ltd., Shanghai, China); acetic acid, lactic acid (Tianjin Institute of Fine Chemical Industry, Tianjin, China).

### 2.2. Instrumentation and Equipment

FlavourSpec^®^ flavor analyzer (G.A.S.); digital vernier calipers (Sata Tool Co., Ltd., Shanghai, China); wine refractometer (ATAGO Co., Ltd., Tokyo, Japan); electronic balance (Cany Precision Instruments Co., Ltd., Shanghai, China); Cary 60 UV-Vis spectrophotometer, high-performance liquid chromatograph (Agilent Technologies Co., Ltd., Waldbronn, Germany); KQ-300E Ultrasonic Cleaner (Kunshan Ultrasonic Instrument Co., Ltd., Kunshan, China); XH-D vortex mixer (Wuxi Wuxin Instrument Co., Ltd., Wuxi, China); Allegra 64 R high-speed freezing centrifuge (Beckman Coulter, Inc., Carlsbad, CA, USA); 50 mL specific gravity bottle with thermometer (Shanghai Xinyi Instrument Factory, Shanghai, China); NH310 High-Quality Portable Computerized Colorimeter (Shenzhen 3NH Technology Co., Ltd., Shenzhen, China).

### 2.3. Methodology

#### 2.3.1. Winemaking

##### Fermentation Flow

The fermentation flow chart for *A. arguta* wine production is shown in Figure 1.

##### Process Stages

(1)Selecting: Selection of high-quality *A. arguta* fruit with uniform ripeness, free of pests and diseases, and free of deterioration for processing.(2)Crushing: We used artificial crushing, causing the fruit to break evenly and added potassium metabisulfite at 0.1 g/L. (Potassium metabisulfite was dissolved with a small amount of distilled water and then added). The crushed fruit was stirred well and then rested for 30 min.(3)Enzymatic processing: To improve the juice yield of *A. arguta*, pectinase was added at 0.1–1%, and the enzyme treatment was carried out at room temperature for 12–24 h. The enzyme was used in the treatment of *A. arguta*.(4)Juicing: The enzymatically processed pulp was pressed to extract the *A. arguta* juice.(5)Adjusting soluble solids: We measured the soluble solid content of the *A. arguta* juice, calculated the amount of added sugar according to the formula below, and adjusted the soluble solid level to 20° Brix.


$$X=\frac{V(20-TSS\times 100)}{\left(100-20\times 0.625\right)}$$


(X: the amount of added sugar; V: the volume of *A. arguta* juice).

(1)Activated yeast: Yeast was added to 5% warm sugar water and placed in a 37 °C constant-temperature water bath for 20 min with stirring to facilitate activation. The activation was complete after foam was abundant.(2)First fermentation: The *A. arguta* juice was added to a 20 L fermenter, and the amount added was not more than 2/3 of the volume of the fermenter. The activated yeast was added at 250 mg/L and stirred well. Three parallel groups were established for each treatment. Fermentation was carried out at (18 ± 2) °C for 7 days. During the fermentation period, the mash was stirred 1–2 times a day, and changes in soluble solids were monitored. At the end of fermentation, the bottom of the mash was removed.(3)Secondary fermentation: The filtered fermentation juice was placed in an airtight chamber at 15 °C for one month for post-fermentation.

#### 2.3.2. Measurement of Basic Physicochemical Indicators

According to GB/T 15038-2006, General Analytical Methods for Wine and Fruit Wine, the soluble solids were determined using a wine refractometer; the alcohol content was determined with an alcohol meter at 25 °C; dry extract content was determined with the density bottle method; and total acid content was determined with the phenolphthalein indicator method [23]. The total sugar content in the wine was determined with anthrone and sulfuric acid colorimetry [36]. The tannin content was determined with the Folin–Denis method, and standard curves for different tannin concentrations were established. The tannin content is expressed as grams of tannin equivalent in a 1 L sample of *A. arguta* wine [27]. The total phenol content was determined with the Folin–Ciocalteu colorimetric method, and standard curves with different gallic acid concentrations were established. The total phenolic content was expressed as milligrams of gallic acid equivalent in 1 L of *A. arguta* wine sample [37]. The total flavonoid content was determined with the AlCl_3_ colorimetric method, and standard curves for different rutin concentrations were established. The total flavonoid content is expressed as milligrams of rutin equivalent in 1 L of *A. arguta* wine sample [37]. We determined the vitamin C content with high-performance liquid chromatography [36].

#### 2.3.3. Measurement of Color

The color of the *A. arguta* wine was evaluated with the *CIEL***a***b** colorimetric standard [38,39]. The color measurement was performed with an NH310 High-Quality Portable Computerized Colorimeter. After a whiteboard calibration, the color index (*L**, *a**, *b**, *C***_ab_*, and *h***_ab_*) of each wine sample was measured, and the color difference value (Δ*E***_ab_*) was calculated. *L** represents the brightness value, and the larger the value of *L**, the larger the brightness; the value of *a** indicates the red–green value (redness when positive, greenness when negative); the value of *b** indicates the yellow–blue value (yellowness when positive, blueness when negative); *C***_ab_* is the saturation; *h***_ab_* is the hue angle; and Δ*E***_ab_* is the total color difference. Each wine sample was measured three times in parallel.
$${{\Delta E}^{*}}_{ab}=[{{\left({\Delta L}^{*}\right)}^{2}+{\left({\Delta a}^{*}\right)}^{2}+{\left({\Delta b}^{*}\right)}^{2}]}^{1/2}$$

#### 2.3.4. Measurement of Organic Acids 

The organic acid content of the *A. arguta* wine was determined by referencing a previously published method [36]. The specific conditions were as follows: The chromatographic column was an Agilent C18-XT column (4.6 mm × 250 mm × 5 mL). The UV detection wavelength was 210 nm, and the injection volume was 10 μL. The mobile phase was an aqueous phosphoric acid solution at pH = 2.3 and methanol (99:1, *v*/*v*). The column temperature was 25 °C, and the flow rate was 0.3 mL/min. The standard curves for the eight organic acids we tested are shown in Table 1.

#### 2.3.5. Measurement of Volatile Composition

The volatile composition of *A. arguta* wine was determined as per Cao et al. [23]. Sample treatment: A 1 mL sample of *A. arguta* wine was placed in a 20 mL headspace injection bottle, and 20 μL of 10 ppm internal standard (4-methyl-2-pentanol) was added. The sample was injected into the sample after incubation at 60 °C for 10 min at 500 r/min. 

There were several reasons for analyzing 4-methyl-2-pentanol as the internal standard. Firstly, 4-methyl-2-pentanol does not exist in *A. arguta* wine. Secondly, 4-methyl-2-pentanol has similar properties to the volatile compounds in *A. arguta* wine, and it does not react chemically with the volatile substances in it. Thirdly, in the experiment, 4-methyl-2-pentanol did not overlap with the volatile compounds in *A. arguta* wine and could be completely separated. The analysis was carried out using 4-methyl-2-pentanol as an internal standard at a concentration of 198 ppb. The volume of the signal peak was 478.01, and the intensity of each peak was approximately 0.414 ppb. The analytical conditions and gas chromatographic conditions are shown in Table 2 and Table 3.

HS-GC-IMS used the NIST library and the IMS library for the qualitative analysis of the substances. The quantitative formula for the content of each volatile compound was calculated as [23]
$$Ci=\frac{Cis*Ai}{Ais}$$

Ci is the calculated mass concentration of any volatile component in µg/L, Cis is the mass concentration of the internal standard (4-Methyl-2-pentanol) used in µg/L, and Ai/Ais is the volume ratio of any signal peak to the signal peak of the internal standard.

#### 2.3.6. Odor Activity Value (OAV) Calculation

The odor activity value (OAV) was used to evaluate the overall aroma contribution of the *A. arguta* wine. The OAV was calculated by dividing the concentration of volatile compounds by the odor threshold. The odor thresholds were determined with reference to the *Compilations of Odour Threshold Values in Air*, *Water and Other Media (Edition 2011)* and values reported in the literature [40,41,42,43,44,45,46,47]. Volatile compounds with an OAV > 1 were considered aroma-active and played an important role in developing aromatic properties.

### 2.4. Sensory Evaluation

The aroma characteristics of the *A. arguta* wines were evaluated with quantitative descriptive analysis (QDA) [23,48,49,50,51]. The sensory evaluation panel consisted of 10 people, including 5 women and 5 men, whose ages ranged from 23 to 53 years, with an average age of 32 years. The sensory evaluation was preceded by 4 weeks of training according to the national standards ISO 6658 and ISO 8586. The first round was designed to familiarize the panel with *A. arguta* wine and the evaluation process and initially generate descriptors; the second round corrected, supplemented, and refined these descriptors with standard objects or corresponding objects and to form a vocabulary of descriptors for the aroma characteristics of *A. arguta* wine, including fruity aroma, floral aroma, fermented aroma, plant and herbal fragrance aroma, and oil aroma; the third round involved scale training and proficiency use training; the fourth round involved comprehensive descriptive guidance and training. The experiments were completed in the wine evaluation room. (The room was appropriately spacious, not too small, but not too large; the noise was limited to 40 dB or less; there was no clutter; there was good ventilation; there were no aromas and concordant odors etc.) Samples of about 20 mL of *A. arguta* wine were presented to each evaluator; each sample was numbered with a 3-digit number and presented to the tasters in a randomized order. The tasters were asked to select and score the aroma characteristics of the samples on a 10-point scale (0 for no odor, 9 for the strongest odor). Scores ranging from 0 to 3 were considered low intensity, 3 to 6 medium intensity, and 6 to 9 high intensity. Each sample was evaluated in triplicate, and the mean value of each sample was represented by the average of the three values based on the 10-point scale.

### 2.5. Statistical Analysis

Excel 2016 was used to statistically organize the experimental data; SPSS version 27.0 was used to perform analysis of variance (ANOVA); statistical analysis of variance of the experimental data was used to check the significant difference of each result; and all the data were expressed as the mean ± SD. Differences between the two groups were considered significant at *p* < 0.05. Simca14.1 was used for OPLS-DA and VIP value analysis. Heatmap analysis, principal component analysis (PCA), and correlation analysis were performed using the omicshare tool (https://www.omicshare.com/tools, accessed on 28 June 2023).

## 3. Results and Discussion

### 3.1. Basic Physicochemical Indicators Analysis

Table 4 shows that there were significant differences between the basic physicochemical indicators of the *A. arguta* wine samples for each variety (*p* < 0.05). The alcohol content of the 10 varieties ranged from 7.17° to 10.07°, which was in line with the provisions of NY/T 1508-2017 ‘Green Food Fruit Wine’ (the alcoholic content of fruit wine ranges from 7° to 18°). ‘Fenglv’ was the highest, followed by ‘Pinglv’ and ‘Tianxinbao’; the residual sugar and solid content determine the type of wine, which can be categorized into dry, semi-dry, semi-sweet, and sweet [52]. The residual sugar content of the ten varieties was less than 4 g/L, corresponding to the dry type. The dry extract content has a certain relationship with the taste of fruit wines, and a too-low dry leachate content results in a bland taste. NY/T 1508-2017 ‘Green Food Fruit Wine’ stipulates that the dry leachate content of fruit wines should be ≥12.0 g/L, and the dry extract content of the 10 varieties we obtained from our testing ranged from 12.25 g/L to 49.59 g/L, which was in line with the brewing standard. The dry extract content of ‘Tianxinbao’ was the highest at 49.59 g/L, which was significantly different from that of the other nine varieties (*p* < 0.05). *A. arguta* fruits are rich in vitamin C and phenolic substances. *A. arguta* can retain a certain amount of vitamin C and phenolic substances after brewing into a fruit wine, which, in turn, can play a role in its anti-oxidant properties, immunity enhancement, and other nutritional and healthcare effects [29]. ‘Kuilv’ had the highest vitamin C and total phenolic content at 0.91 g/L and 0.82 g/L, respectively, which differed significantly from the other varieties (*p* < 0.05); ‘Jialv’ had the highest total flavonoid content at 0.48 g/L, which differed significantly from the other varieties (*p* < 0.05), and the total flavonoid content was the lowest in ‘Cuiyu’ at 0.20 g/L. Tannins are an important source of bitterness and astringency in fruit wines, as well as an important constituent of their structure, backbone, and taste, and they have a positive effect on color stability, oxidation prevention, and off-flavor removal [53,54]. ‘Cuiyu’, ‘Longcheng No.2’, and ‘Tianxinbao’ had higher tannin contents at 4.63 g/L, 4.46 g/L, and 4.28 g/L, respectively, which were significantly higher than the other seven varieties (*p* < 0.05).

### 3.2. CIELab Parameters Analysis

Chromaticity is one of the important bases for evaluating the quality of fruit wines and is closely related to consumer acceptance. Table 5 shows that there were significant differences (*p* < 0.05) in the color of the *A. arguta* wine samples made with different varieties. *L** reflects the brightness of the fruit wines. The largest *L** values were 31.65 for ‘Kuilv’ and ‘Longcheng No.2’, with the brightest color, followed by ‘Cuiyu’ and ‘Wanlv’. The smallest *L** value was 30.14 for ‘Pinglv’, with the darkest color. The chroma *a** value reflects the red and green degree of a fruit wine color. The *a** values of the 10 samples were all positive, with the larger positive value indicating a higher degree of redness. The *a** values ranged from 2.34 to 2.88, with the largest being ‘Pinglv’ and the smallest being ‘Wanlv’. The chroma *b** values reflect the yellow and blue degree of fruit wines. The *b** values of the 10 samples were all positive, with the larger positive value indicating that the color was more yellow. The largest *b** value was 5.08 for ‘Kuilv’, followed by ‘Wanlv’ and ‘Longcheng No.2, and the smallest *b** value was 3.15 for ‘Pinglv’. The saturation *C***_ab_* value is a combination of *a** and *b**, reflecting the color fullness of fruit wines. The highest saturation was for ‘Kuilv’, followed by ‘Wanlv’, and the lowest was for ‘Pinglv’ at 4.27. The *h**_ab_ value indicates the hue angle. A hue angle between 0° and 90° indicates a red-to-yellow tone, between 90° and 180° indicates a yellow-to-green tone, between 180° and 270° indicates a green-to-blue tone, and between 270° and 360° indicates a blue-to-red tone [55]. The hue angle of the 10 wine samples measured in the experiment ranged from 47.59 to 65.19, indicating that the hue values of each group of samples were between the red and yellow tones.

Δ*E***_ab_* values were calculated for the ten samples, using the ‘Kuilv’ sample with the highest *L** value as the base value. They were calculated according to the range of color difference units NBS (National Bureau of Standards units) given by the CIE 1976 (*L***a***b**) color space system to describe the degree of color difference between wine samples [23]. The values in Table 5 show that the color difference between ‘Kuilv’ and the other nine varieties was small.

### 3.3. Organic Acids Analysis

Organic acids play a crucial role in the flavor of fruit wines, affecting their chemical stability and pH, and are closely related to the quality of the wine [56,57]. Table 6 shows that there were significant differences in the organic acid content of the wine samples (*p* < 0.05), with the main organic acids being citric, quinic, and malic acid.

Citric acid has a refreshing sour flavor, a mild and crisp taste, freshness, and a short aftertaste. The highest citric acid content was found in ‘Longcheng No.2’ with 7.45 g/L, which was significantly higher than the other nine varieties (*p* < 0.05), followed by ‘Kuilv’, and the lowest citric acid content was found in ‘Pinglv’ at 3.12 g/L. The highest quinic acid content was found in ‘Longcheng No.2’ at 7.04 g/L, which was significantly higher than in the other nine varieties (*p* < 0.05). This was followed by ‘Xinlv’, and the lowest quinic acid content was found ‘Pinglv’ at 2.82 g/L. Malic acid is 20% more acidic than citric acid, has a soft flavor, a distinct aroma, a bitter astringent taste, and a longer presentation time. Malic acid is more abundant in *A. arguta* at the end of alcoholic fermentation, and its content gradually decreases after malolactic fermentation (MLF), which is a very important secondary fermentation process in the production of fruit wines [35,58]. Usually, malolactic fermentation occurs after the completion of alcoholic fermentation, where lactic acid bacteria convert the sharp L-malic acid into the softer L-lactic acid, which results in a lower acidity of fruit wines and a more stabilized body [59]. The highest malic acid content was 2.92 g/L in ‘Tianxinbao’, followed by ‘Fenglv’, and the lowest malic acid content was 0.83 g/L in ‘Kuilv’. The highest lactic acid content was 0.8 g/L for ‘Tianxinbao’, followed by ‘Xinlv’, and the lowest lactic acid content was in ‘Jialv’. The highest acetic acid content was in ‘Fenglv’, followed by ‘Xinlv’, and the lowest acetic acid content was in ‘Wanlv’. Mangiferic acid was detected in ‘Kuilv’, ‘Jialv’, ‘Wanlv’, ‘Lvbao’, ‘Tianxibao’, and ‘Longcheng No.2’. The highest shikimic acid content was detected in ‘Wanlv’ at 0.06 g/L, which was significantly higher than in the other varieties (*p* < 0.05). Succinic acid was detected in ‘Kuilv’, ‘Jialv’, ‘Wanlv’, ‘Pinglv’, and ‘Longcheng No.2’. The highest succinic acid content was detected in ‘Kuilv’ at 2.41 g/L, which was significantly higher than in the other varieties (*p* < 0.05).

Cluster analysis results can better reflect the characteristics of organic acid substances in the different *A. arguta* wine samples (Figure 2). Based on the organic acid cluster analysis of each variety, it can be seen that the ten *A. arguta* wine samples can be classified into three categories when the value of the transverse tangent line is taken as four. The first category includes ‘Longcheng No.2’; the second category includes ‘Tianxinbao’, ‘Jialv’, and ‘Pinglv’; and the third category includes ‘Kuilv’, ‘Xinlv’, ‘Cuiyu’, ‘Fenglv’, ‘Wanlv’, and ‘Lvbao’, indicating that when the crosscut line takes a value of four, each class contains samples similar organic acids. Thus, the results also better clustered the different types of *A. arguta* wines together.

### 3.4. HS-GC-IMS Analysis of Different A. arguta Wines

The volatile flavor substances of fruit wines mainly come from the fruit itself, yeast fermentation, and aging processes [60]. Aroma substances are indispensable indicators for evaluating the quality of fruit wines, and together, these volatile flavor substances, through the enrichment effect, provide *A. arguta* wines with a rich and unique flavor. Headspace gas chromatography–ion mobility spectrometry (HS-GC-IMS) is a commonly used method for separating and quantifying aroma substances in food.

To analyze the differences in volatile flavor compounds in different varieties of *A. arguta* wine, the fingerprints of these volatile flavor compounds were constructed based on all the signal peaks in the two-dimensional HS-GC-IMS spectra. Each wine sample was measured three times; the darker the color, the greater the peak intensity and the higher the content. The fingerprints showed the composition of and differences in the volatile flavor compounds in the samples. Figure 3 shows that ‘Kuilv’ had a high content of diethyl acetal M, acetic acid D, etc.; ‘Fenglv’ had a high content of heptan-2-ol, isobutyl acetate (E)-3-nonen-2-one, etc.; ‘Jialv’ had high a content of propyl propanoate, hexyl acetate, ethyl hexanoate, etc.; ‘Wanlv’ had a high content of 1-octen-3-ol, methyl 2-methylbutanoate, ethyl octanoate M ethyl octanoate D, etc.; ‘Xinlv’ had a high content of 1-octen-3-ol, ethyl crotonate butanal, etc.; ‘Lvbao’ had a high content of ethyl propanoate, etc.; ‘Tianxinbao’ had a high content of propanol, 2-methylbutanal, 1-hexanol M, methyl 2-methylbutanoate, butanal, ethyl pentanoate, ethyl formate, 2-pentanone, octanal, ethyl heptanoate, etc.; ‘Longcheng No.2’ had a high content of diethyl acetal D, isobutyl butyrate, ethyl isobutyrate, etc.

Figure 4 shows the two-dimensional spectrum created by the Reporter plugin. The horizontal coordinates in the two-dimensional spectrum indicate the relative drift time (RIP, unitless); the vertical coordinates indicate the GC retention time(s); and the volatile substance content is represented by the color depth, with the darker color indicating a greater concentration of the substance. The results show that the volatile components of different varieties of *A. arguta* wines could be separated using GC within 30 min, and the types of compounds detected were more or less the same. However, there were some differences in the volatile contents in different wine samples, indicating that the differences in the proportional compositions of the volatiles were one of the key material bases for differences in the stylistic characteristics of different *A. arguta* wines.

Taking the ‘Kuilv’ variety as a reference, the rest of the spectrum is subtracted from the signal peaks in ‘Kuilv’ to obtain the difference spectrum (Figure 5). The blue area indicates that the amount of substance in a sample is lower than in ‘Kuilv’, and the red area indicates that the amount of substance in a sample is higher than in ‘Kuilv’. Again, the darker the color, the greater the difference. The difference spectrum in Figure 6 shows that the ethyl isovalerate, diethyl acetal M, and acetic acid D contents were higher in ‘Kuilv’ than in other varieties.

### 3.5. Analysis of the Differences in the Volatile Compositions of Different A. arguta Wines

Aroma is one of the most important sensory characteristics of fruit wines [23]. The volatile compounds in the *A. arguta* wine samples were analyzed qualitatively and quantitatively using VOCal to view the analytical spectra and data. In total, 51 volatile compounds were detected and identified by using the built-in NIST and IMS databases of HS-GC-IMS, and the most commonly detected volatile compounds were 24 ester compounds, 12 alcohols, 9 aldehydes, 3 ketones, 2 terpenes, and 1 acid (Table 7). The types of volatile compounds detected in the ten varieties were the same but with significant differences in content. Perhaps because this study used the same fermentation methods, enzymes and yeasts, the volatile profiles of the different *A. arguta* wines were similar. ‘Fenglv’ had the highest content of volatile compound, measured at 43,987.49 mg/L, followed by ‘Pinglv’ at 43,382.33 mg/L, ‘Cuiyu’ at 43,150.77 mg/L, ‘Xinlv’ at 42,475.51 mg/L, ‘Lvbao’ at 42,397.46 mg/L, ‘Longcheng No.2’ at 42,100.46 mg/L, ‘Jialv’ at 42,078.03 mg/L, ‘Kuilv’ at 42,021.49 mg/L, ‘Tianxinbao’ at 41,850.43 mg/L, and ‘Wanlv’ at 40,288.58 mg/L. Considering the proportions of each type of volatile compound in each variety, alcohols accounted for the largest proportion (57.18–60.98%), followed by esters (32.22–35.65%). Alcohols and esters were the main aroma compounds in the ten samples. Park et al. [35] also found that esters and alcohols were the main aroma components of *A. arguta* wine, which was consistent with our results.

#### 3.5.1. Alcohols

Alcohol compounds are important contributors to wine flavor and are the main products derived from yeast during alcoholic fermentation via sugar catabolism or amino acid decarboxylation and deamidation [52]. Table 7 shows the highest alcohol content detected in the *A. arguta* wine was in ‘Fenglv’ at 26,053.39 mg/L, followed by ‘Cuiyu’ at 25,931.00 mg/L, ‘Longcheng No.2’ at 25,671.24 mg/L, ‘Pinglv’ at 25,656.39 mg/L, ‘Lvbao’ at 25,162.66 mg/L, ‘Kuilv’ at 24,583.40 mg/L, ‘Tianxinbao’ at 24,539.92 mg/L, ‘Xinlv’ at 24,459.99 mg/L, ‘Jialv’ at 24,232.49 mg/L, and ‘Wanlv’ at 23,038.88 mg/L. Alcohols accounted for the largest proportion of components in each variety. Alcohols were previously reported to contribute at least 65% of the aroma characteristics of *A. arguta* wine [33]. The alcohol aroma component is mainly composed of alcohol and brandy flavors, which can give the wine a sense of complexity. Therefore, ‘Fenglv’ has a strong mellow feel.

#### 3.5.2. Esters

It was reported that esters are the most abundant type of volatile compounds in *A. arguta* wines [24,33,35]. Table 7 shows that a total of 24 esters were detected in the *A. arguta* wine samples, with the highest content in ‘Fenglv’ at 15,127.74 mg/L, followed by ‘Jialv’ at 15,074.84 mg/L, ‘Xinlv’ at 15,027.73 mg/L, ‘Kuilv’ at 14,981.45 mg/L, ‘Pinglv’ at 14,735.32 mg/L, ‘Lvbao’ at 14,363.20 mg/L, ‘Cuiyu’ at 14,231.72 mg/L, ‘Wanlv’ at 14,198.27 mg/L, ‘Tianxinbao’ at 13,903.61 mg/L, and ‘Longcheng No.2’ at 13,566.64 mg/L. The most abundant esters in the ten wines in this study were ethyl acetate, isoamyl acetate, ethyl isovalerate, ethyl hexanoate, and ethyl butanoate. Ethyl butyrate was previously reported to be the predominant aroma component of mature *A. arguta* fruits [61] and was also detected in this study. These esters provide apple, pineapple, banana, strawberry, cheese, sour, brandy, and floral flavors [23,33,35,62].

#### 3.5.3. Aldehydes

Generally, aldehyde compounds are formed via alcohol oxidation [24]. Aldehydes play catalytic roles in wines and are the main source of aroma. A suitable amount of aldehyde in wine is necessary to keep the taste free of pungency [23,58,63]. In this study, the aldehyde content of the varieties ranged from 4.72 to 5.99%, with the highest content in ‘Tianxinbao’ at 2505.63 mg/L, followed by ‘Xinlv’ at 2388.27 mg/L, ‘Wanlv’ at 2378.93 mg/L, ‘Pinglv’ at 2356.28 mg/L, ‘Cuiyu’ at 2344.04 mg/L, ‘Lvbao’ at 2311.42 mg/L, ‘Longcheng No.2’ at 2276.03 mg/L, and ‘Kuilv’ at 1983.70 mg/L. 2-methylbutanal, butanal, pentanal, and diethyl acetal positively affect the flavor of *A. arguta* wines, even at low concentrations, and alter that flavor along with other volatile aromas [63].

#### 3.5.4. Ketones

Ketones were present at only 0.40–0.95% of the total content of the wine samples. Three ketones were detected: (E)-3-nonen-2-one, 2-pentanone, and 4-methyl-2-pentanone. These ketones synergize with the wine to present a pleasant aromatic flavor and are not harmful to humans [24,35,58].

#### 3.5.5. Acids

Acids were present at only 0.28–0.37% of the total content of the wine samples. Acids are important volatile components that affect the complexity and fruitiness of wine [58]. Acetic acid was detected in the ten wine samples. Acetic acid has also been detected in cider [64], and is mainly a by-product of yeast metabolism during fermentation [65].

#### 3.5.6. Terpenoids

Terpenoids were present at only 0.34–0.83% of the total content of the wine samples. Terpenoids determine the typical aroma profile of wines, albeit at low levels and usually in the form of glycosidic binding [66]. In this study, two terpenoids were detected, i.e., terpinolene and styrene.

### 3.6. Multivariate Statistical Analysis

#### 3.6.1. Principal Component Analysis (PCA)

Principal component analysis (PCA) is a multivariate statistical analysis technique. Many complex and hard-to-find variables in the original sample are detected by identifying several principal component factors. Then, regularities and differences between samples are assessed based on the contributions of the principal component factors of different samples [23,26]. The volatile compounds identified using HS-GC-IMS were analyzed with PCA (Figure 6). The results show that the ten *A. arguta* wine samples were well differentiated according to their aroma characteristics and varieties. The contribution rate of PC1 was 27.8% and that of PC2 was 19.9%, and the ten groups of samples showed a clear trend of separation on the two-dimensional plot, with no outlier samples in the same kinds of *A. arguta* wine. The samples clustered well. There were significant differences in the overall aroma substances of the ten groups, and they were distinguished. Figure 6 shows that ‘Pinglv’, ‘Cuiyu’, ‘Xinlv’, ‘Lvbao’, ‘Longcheng No.2’, and ‘Wanlv’ were closer together; ‘Jialv’ and ‘Kuilv’ were closer together; and ‘Fenglv’ and ‘Tianxinbao’ were farther apart, indicating that there were significant differences in the aroma characteristics of the different samples.

#### 3.6.2. Hierarchical Cluster Analysis

To further clarify the affinity of the volatile flavor profiles of the different samples, a hierarchical clustering heatmap was drawn based on the concentration data of the volatile compounds (Figure 7). The figure shows that in terms of volatile flavor substance composition, ‘Tianxinbao’ and ‘Fenglv’ were each categorized separately; ‘Kuilv’ and ‘Jialv’ clustered into one category; and ‘Wanlv’, ‘Lvbao’, ‘Xinlv’, ‘Pinglv’, ‘Longcheng No.2’, and ‘Cuiyu’ clustered into one group. The results of the hierarchical cluster analysis were consistent with those of the principal component analysis. This indicated that ‘Kuilv’ and ‘Jialv’ were more similar in terms of flavor substance composition, and ‘Wanlv’, ‘Lvbao’, ‘Xinlv’, ‘Pinglv’, ‘Longcheng No.2’, and ‘Cuiyu’ were similar in terms of the composition of flavor substances. The flavor compositions of ‘Tianxinbao’ and ‘Fenglv’ were unique. The main reason for the differences between the samples was the presence of several flavor components that were significantly higher in the ‘Tianxinbao’ and ‘Fenglv’ samples than in the other samples. In terms of specific volatile substances, there was some similarity in the content distribution of compounds in the 10 samples, with 2-methyl-1-propanol, 3-methyl-1-butanol D, 4methyl-1-pentanol, ethanol, ethyl isovalerate, butyl pentanoate, ethyl acetate, ethyl hexanoate, and isopentyl propanoate being the most common components with concentration advantages in different *A. arguta* wines (Table 7). However, each variety of *A. arguta* wine had unique components with higher contents. Figure 7 shows that ‘Tianxinbao’ had higher compound contents of propanol, octanal, 1-hexanol M, 2-pentanone, 2-methylbutanal, ethyl heptanoate, and heptanal, while ‘Fenglv’ had higher contents of heptan-2-ol, isobutyl acetate, isopentyl propanoate, 2-methyl-1-propanol, ethyl isovalerate, 3-methyl-1-butanol M, (E)-3-nonen-2-one, and butyl butanoate.

#### 3.6.3. OPLS-DA Analysis

OPLS-DA is a supervised statistical method for discriminant analysis that identifies sample differences and obtains characteristic markers from sample differences [23,40]. Based on the content data of each aroma substance, OPLS-DA analysis was performed (Figure 8A). A total of 12 variance variables that contributed more to the model were screened out using the criteria VIP > 1 and *p <* 0.05. These variables were ethyl butanoate, acetic acid D, methyl acetate, heptanal, butyl butanoate, ethyl acetate, ethyl crotonate, ethyl isobutyrate, 1-octen-3-ol, ethyl isovalerate, 2-methylbutanal, and ethyl hexanoate.

### 3.7. Characterization of Flavor Profiles and Differential Aroma Compounds of Different Varieties of A. arguta Wine

The influence of volatile flavor compounds on the formation of flavor characteristics depends not only on the level of their content but also correlates with the flavor threshold of the compounds [40]. Based on the theory of odor contribution, the flavor characteristics of different *A. arguta* wines were characterized and the key odor compounds responsible for the differences in the stylistic features of different *A. arguta* wines were revealed. The aroma activity values of each volatile compound are shown in Table 8. A total of 18 volatile aroma compounds with OAVs > 1 were detected in the 10 *A. arguta* wine samples. The esters contained the most aromatic compounds with OAVs > 1 for 10 species: ethyl isovalerate, butyl butanoate, ethyl butanoate, ethyl crotonate, ethyl hexanoate, ethyl isobutyrate, ethyl pentanoate, isoamyl acetate, isobutyl acetate, ethyl octanoate D, and ethyl octanoate M; the two alcohols were 1-octen-3-ol and 4-methyl-1-pentanol; the four aldehydes were 2-methylbutanal, butanal, octanal, and pentanal; and the terpene was terpinolene. Although the OAVs of the key compounds in the ten samples varied, on the whole, the OAVs of the esters were higher than those of other kinds of compounds. Among the ester compounds, the OAV of ethyl crotonate was the highest at 466.37–726.27, which made a greater contribution to the overall aroma; ester compounds were dominated by fruity and floral aromas, which also indicated that esters were the most important contributors to the overall aroma. 

Hierarchical analysis was used to cluster the volatile aroma substances of the ten *A. arguta* wine samples with OAVs > 1. In the heatmap analysis of each variety (Figure 9), the red color indicates that the volatile compound was highly expressed in the samples, and the blue color indicates that the volatile compound was expressed at a lower level. The volatile aroma substance contents with OAVs > 1 varied greatly between the wine samples.

To further clarify the key components contributing to differences in the style characteristics of different *A. arguta* wines, a partial least squares discriminant analysis was performed based on volatile components with an OAV > 1 (Figure 8B). The results show that ethyl butanoate, ethyl pentanoate, ethyl crotonate, ethyl isobutyrate, butyl butanoate, 2-methylbutanal, ethyl isovalerate, and ethyl hexanoate were the main contributing components.

### 3.8. Sensory Evaluation Characteristics of A. arguta Wine

The *A. arguta* wine samples were clear, transparent, and lustrous. ‘Pinglv’ was more oxidized and, therefore, its samples were darker in color, while the other varieties were pale yellow (Figure 10). The organoleptic evaluation was carried out by using the descriptors ‘fruity aroma’, ‘floral aroma’, ‘plant and herb fragrance aroma’, ‘fermented aroma’, ‘oily aroma’, and ‘total aroma’ to evaluate the aroma profile of the *A. arguta* wines. QDA is a descriptive analytical test that combines qualitative and quantitative methods to analyze sensory data using statistical methods [65]. Statistical analyses showed that the samples differed in each descriptor (Figure 11). These significant differences indicated that the flavor intensity of each sample was significantly different. The ‘fruity’ and ‘floral’ aromas are the most fundamental parts of *A. arguta* wine’s flavor. Therefore, these two aroma characteristics are important indicators of aroma quality. As can be seen from the GC-IMS results, esters provided the largest variety of volatile flavor substances. Esters contribute to the desirable fruity sensory profile of *A. arguta* wines. ‘Fenglv’ had a ‘fruitier’ flavor compared with the other samples, which was consistent with the GC-IMS results. Of the ten *A. arguta* wines, ‘Fenglv’ had the highest total ester content. ‘Tianxibao’ showed higher floral aromas. In addition, ‘Tianxibao’ also showed a high botanical and herbal aroma, and butyraldehyde, octanal, pentanal, and glutaraldehyde may be closely related to the description of that aroma, i.e., aldehydes and alcohols are usually associated with ‘green’, ‘botanical’, etc. The total ester compounds in the GC-IMS were the highest of all the wines. In the GC-IMS results, nine aldehydes were detected, with ‘Tianxibao’ having the highest aldehyde content at 2505.63 mg/L. Fermentation flavors are mainly produced in the fermentation and aging stages. All the varieties of *A. arguta* wines had a low oil aroma. The total aromas, from highest to lowest, were ‘Tianxibao’, ‘Fenglv’, ‘Xinlv’, ‘Wanlv’, ‘Jialv’, ‘Cuiyu’, Kuilv’, ‘Lvbao’, ‘Longcheng No.2’, and ‘Pinglv’. The OPLS-DA analysis screened eight key volatiles, mainly esters, in the aroma substances of *A. arguta* wines based on OAVs > 1 in the volatile fractions of the different samples, whose aroma profiles were mainly fruity and floral. Meanwhile, in the correlation analysis between the key volatile substances and the sensory flavors (Figure 12), the red boxes have positive Pearson correlation coefficients. The results show that the esters such as ethyl butanoate, ethyl pentanoate, ethyl crotonate, butyl butanoate, ethyl isovalerate, and ethyl hexanoate were positively correlated with ‘fruity’ and ‘floral’ aromas, and all the key aroma substances were positively correlated with the overall aroma of *A. arguta* wine. This suggests that the eight compounds with VIP > 1 based on OAVs may influence the differences between groups.

## 4. Conclusions

Modern research has concluded that raw materials are the main determinants of wine and fruit wine quality. The selection of varieties is particularly important. The selection of raw material varieties for winemaking should not only pay attention to the nutritional value of fruit wines, but also comprehensively consider the color, flavor, and other indicators. In this study, the basic physicochemical indicators, color, organic acids, volatile components, and sensory quality of original wines made from 10 *Actinidia arguta* varieties were examined and analyzed, and the results showed that there were significant differences in the quality of wines from different varieties. Of these, ‘Kuilv’ had the best vitamin C content, total phenol content, and color; ‘Jialv’ had the highest total flavonoid content, followed by ‘Fenglv’; ‘Tianxinbao’ had the highest dry extract content; and ‘Fenglv’ had the highest volatile flavor substance content. Sensory evaluation showed that ‘Tianxinbao’ had the highest total aroma score, followed by ‘Fenglv’. The comprehensive analysis reveals that ‘Kuilv’, ‘Fenglv’, and ‘Tianxinbao’ were more suitable for winemaking.

Headspace gas chromatography–ion mobility spectrometry was used to determine the volatile components in different varieties of *Actinidia arguta* wines. A total of 51 volatile compounds were identified, including 24 esters, 12 alcohols, 9 aldehydes, 3 ketones, 2 terpenes, and 1 acid. The odor activity values were calculated, and a total of 18 volatile compounds with odor activity values > 1 were screened out. The odor activity values of compounds with odor activity values >1 in the composition of *Actinidia arguta* wine samples from different varieties were used as Y variables for orthogonal partial least squares discriminant analysis to obtain characteristic markers of sample variation. The results show that the compounds causing differences in the aroma could include ethyl butyrate, ethyl valerate, ethyl crotonate, ethyl isobutyrate, butyl butyrate, 2-methylbutyraldehyde, ethyl isovalerate, and ethyl hexanoate. The headspace gas chromatography–ion mobility spectrometry method can show the commonalities and differences between samples, which can make up for the insufficiency of manual sensory analysis and play a useful complementary role in the quality evaluation of actinomycete wines. This can provide technical references for the rapid identification of actinomycetes wines, the selection of brewing varieties, and quality evaluation. However, the IMS database was not complete enough, which made it impossible to identify some of the detected volatiles. Therefore, the gradual enrichment of the database should be the main focus for development of the IMS database in the future.

## Figures and Tables

**Figure 1 foods-12-03345-f001:**
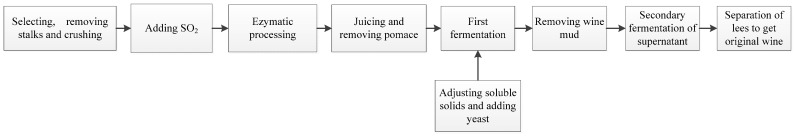
Fermentation flow chart for *A. arguta* wine production.

**Figure 2 foods-12-03345-f002:**
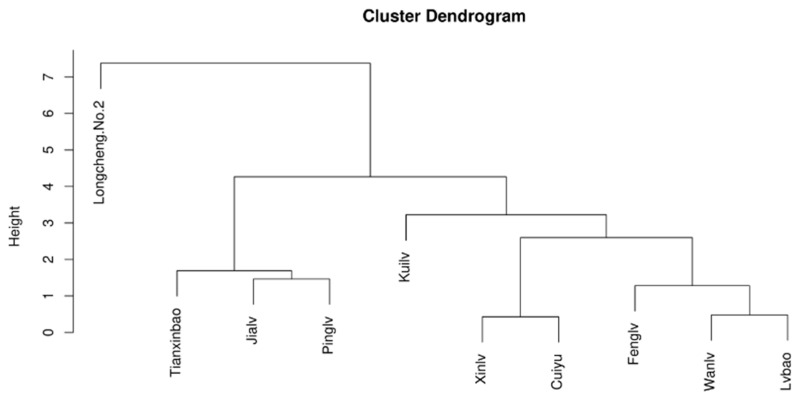
Cluster analysis of organic acids of ten *A. arguta* wine samples.

**Figure 3 foods-12-03345-f003:**
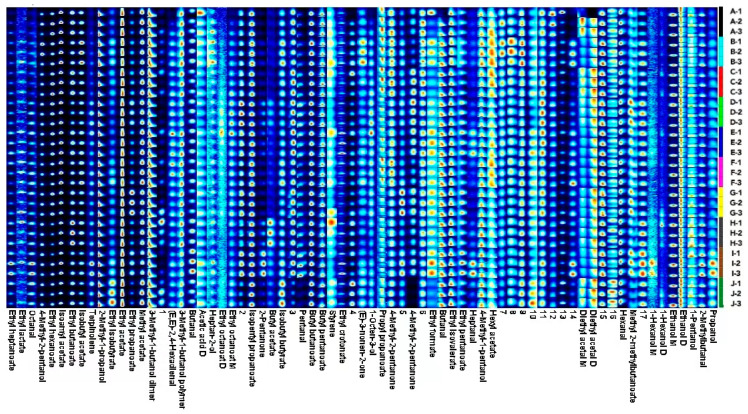
Fingerprints of volatile compounds in ten *A. arguta* wine samples. Note: The letters A to J refer to ‘Kuilv’, ‘Fenglv’, ‘Jialv’, ‘Wanlv’, ‘Xinlv’, ‘Pinglv’, ‘Lvbao’, ‘Cuiyu’, ‘Tianxinbao’, and ‘Longcheng No.2’, respectively.

**Figure 4 foods-12-03345-f004:**
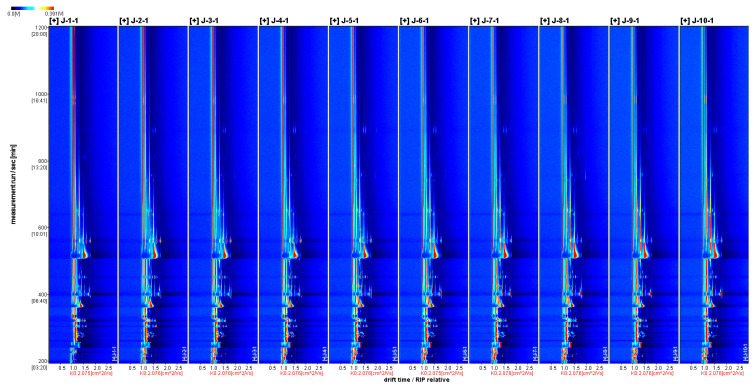
Two-dimensional spectrum of ten *A. arguta* wine samples. Note: The numbers from 1 to 10 refer to ‘Kuilv’, ‘Fenglv’, ‘Jialv’, ‘Wanlv’, ‘Xinlv’, ‘Pinglv’, ‘Lvbao’, ‘Cuiyu’, ‘Tianxinbao’, and ‘Longcheng No.2’, respectively.

**Figure 5 foods-12-03345-f005:**
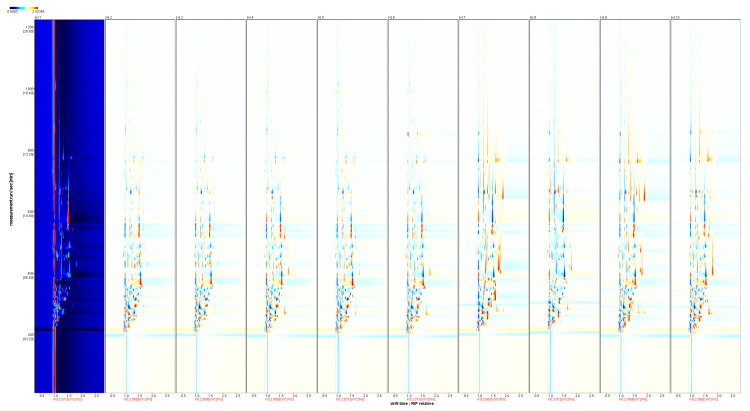
HS-GC-IMS difference spectrum of ten *A. arguta* wine samples. Note: Numbers from 1 to 10 refer to ‘Kuilv’, ‘Fenglv’, ‘Jialv’, ‘Wanlv’, ‘Xinlv’, ‘Pinglv’, ‘Lvbao’, ‘Cuiyu’, ‘Tianxinbao’, and ‘Longcheng No.2’, respectively.

**Figure 6 foods-12-03345-f006:**
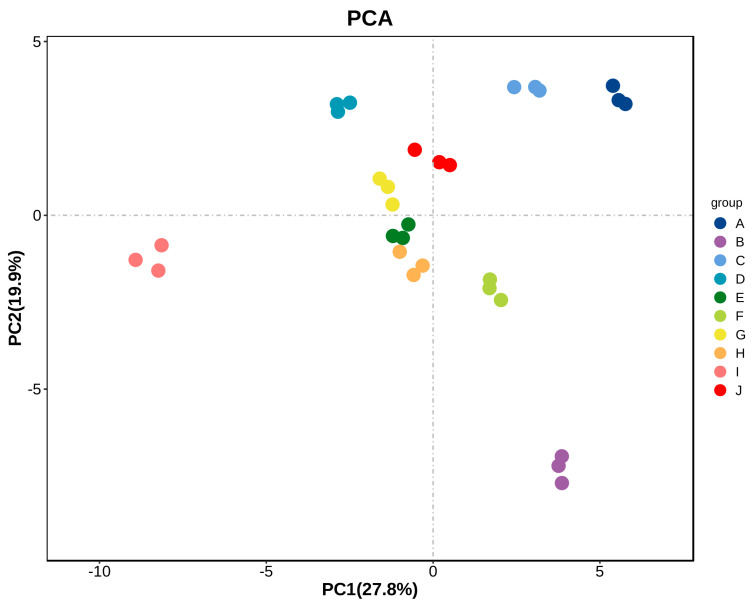
PCA analysis of aromatic compounds of ten *A. arguta* wine samples. Note: Letters from A to J refer to ‘Kuilv’, ‘Fenglv’, ‘Jialv’, ‘Wanlv’, ‘Xinlv’, ‘Pinglv’, ‘Lvbao’, ‘Cuiiyu’, ‘Tianxinbao’ and ‘Longcheng No.2’, respectively.

**Figure 7 foods-12-03345-f007:**
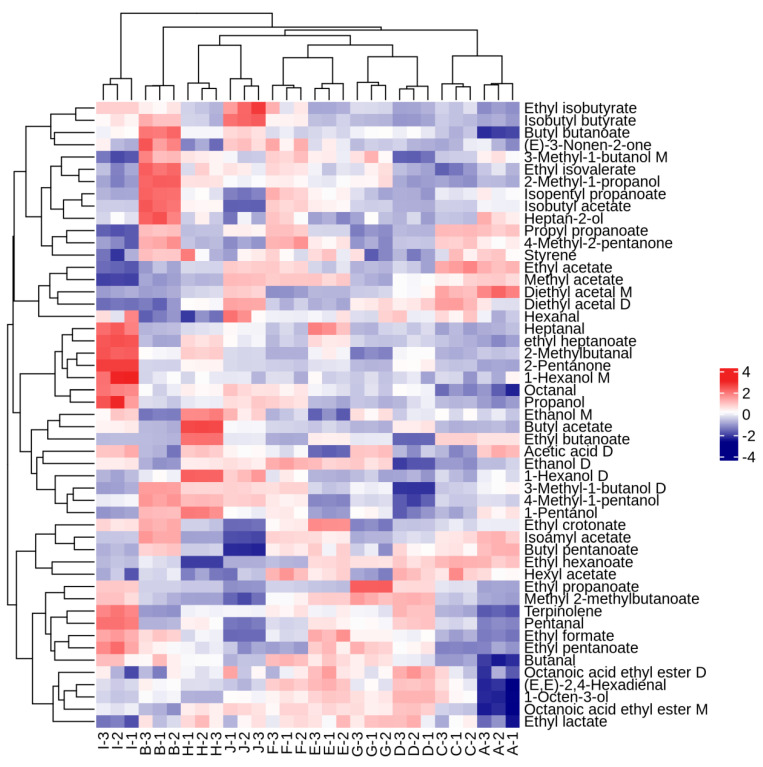
Heatmap analysis of aromatic compounds of ten *A. arguta* wine samples. Note: Letters from A to J refer to ‘Kuilv’, ‘Fenglv’, ‘Jialv’, ‘Wanlv’, ‘Xinlv’, ‘Pinglv’, ‘Lvbao’, ‘Cuiiyu’, ‘Tianxinbao’ and ‘Longcheng No.2’, respectively.

**Figure 8 foods-12-03345-f008:**
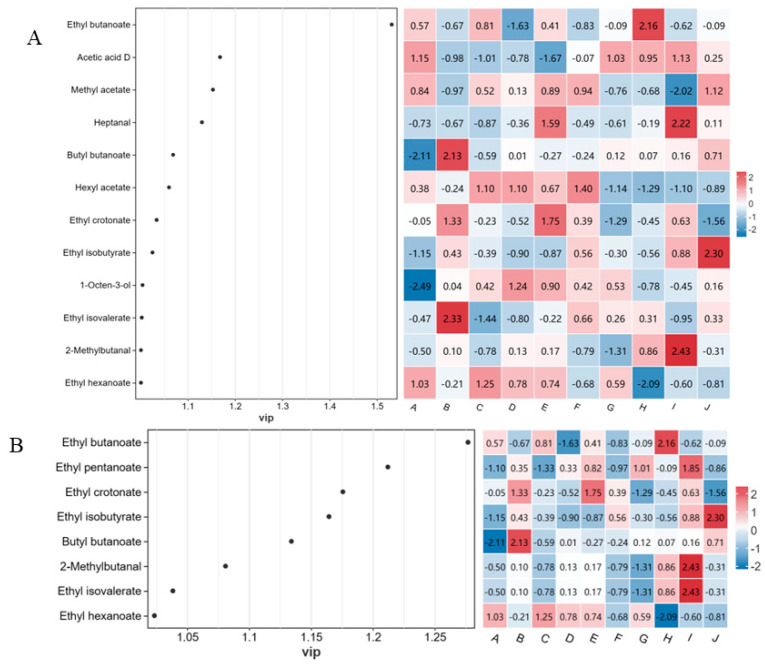
OPLS-DA analysis VIP maps of *A. arguta* wine based on aromatic compounds matter concentration (**A**) and odor activity value (OAV) (**B**). Note: Letters from A to J refer to ‘Kuilv’, ‘Fenglv’, ‘Jialv’, ‘Wanlv’, ‘Xinlv’, ‘Pinglv’, ‘Lvbao’, ‘Cuiiyu’, ‘Tianxinbao’ and ‘Longcheng No.2’, respectively.

**Figure 9 foods-12-03345-f009:**
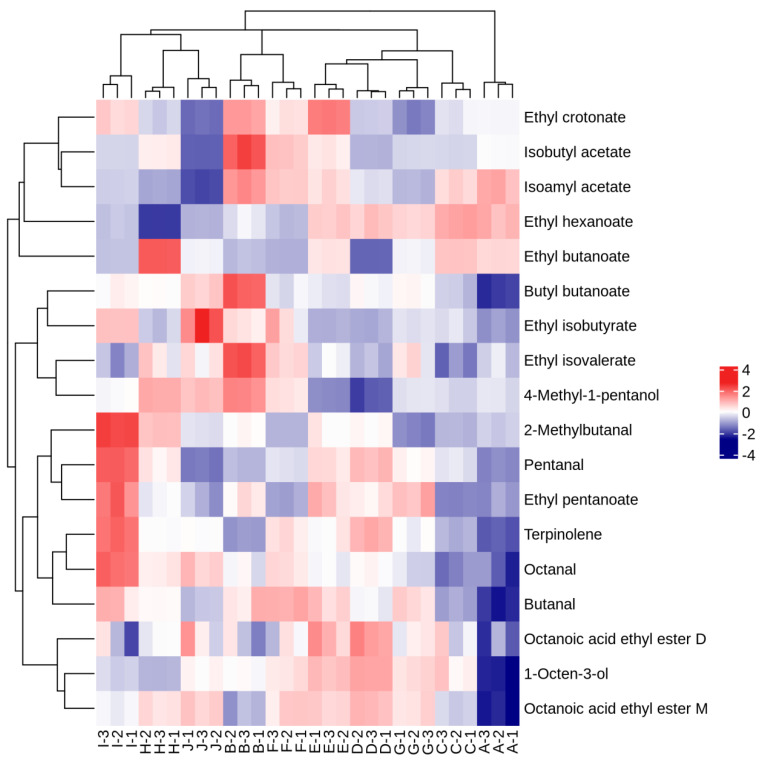
Heatmap analysis of volatile aroma compounds with OAVs > 1 in ten *A. arguta* wine samples.

**Figure 10 foods-12-03345-f010:**
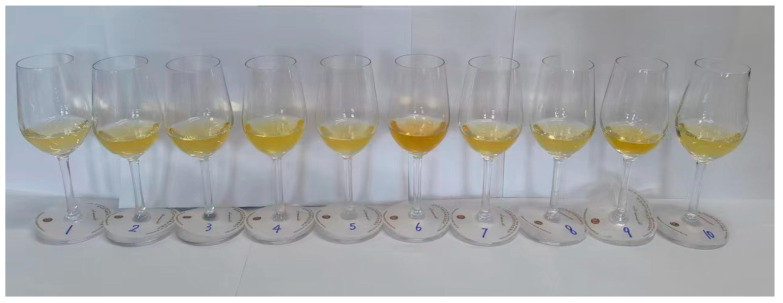
Picture of ten *A. arguta* wine samples. Note: Numbers from 1 to 10 refer to ‘Kuilv’, ‘Fenglv’, ‘Jialv’, ‘Wanlv’, ‘Xinlv’, ‘Pinglv’, ‘Lvbao’, ‘Cuiyu’, ‘Tianxinbao’, and ‘Longcheng No.2’, respectively.

**Figure 11 foods-12-03345-f011:**
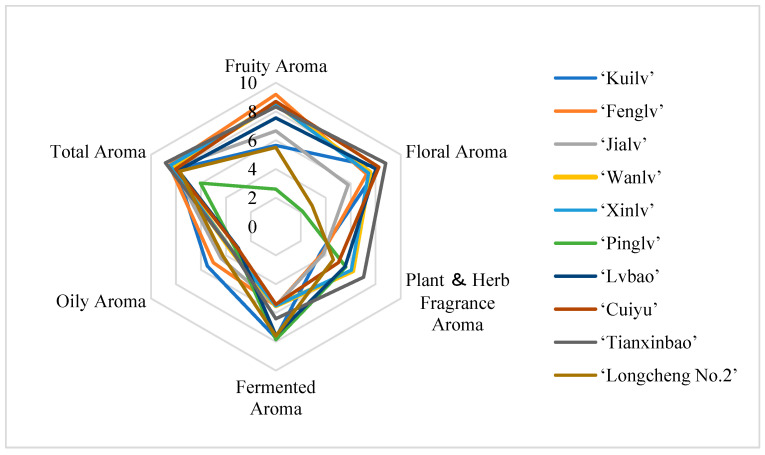
Description of the quantitative aroma analysis of ten *A. arguta* wine samples.

**Figure 12 foods-12-03345-f012:**
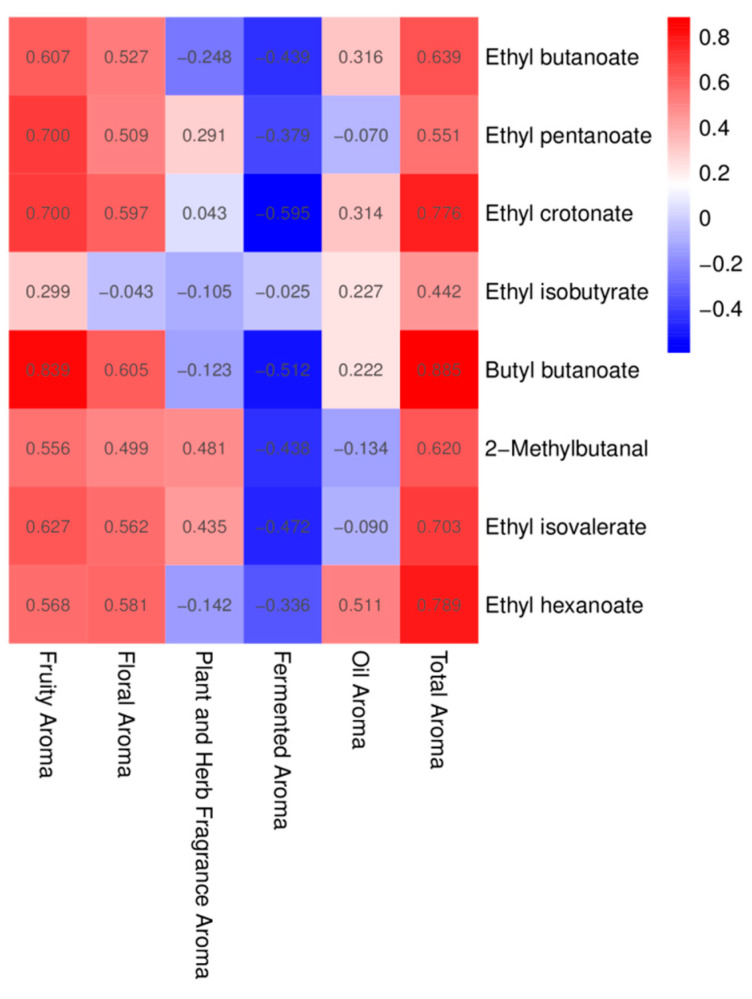
Analysis of the correlation between the key aroma compounds and organoleptic flavors of ten *A. arguta* wine samples.

**Table 1 foods-12-03345-t001:** Organic acid standard curves.

Name	Standard Curves	R^2^
Oxalic Acid	f(x) = 15,184x + 28.758	0.9994
Malic Acid	f(x) = 917.66x + 19.027	0.9995
Shikimic Acid	f(x) = 65,935x + 474.72	0.9995
Lactic Acid	f(x) = 683.52x − 60.731	0.9997
Acetic Acid	f(x) = 749.42x + 3.8359	0.999
Citric Acid	f(x) = 1487x − 4.2267	0.9998
Succinic Acid	f(x) = 752.1x + 1.774	0.9995
Quinic Acid	f(x) = 611.85x − 0.598	0.9998

**Table 2 foods-12-03345-t002:** Analysis conditions for the gas-ion transport spectral unit and automatic headspace injection unit.

Gas-Ion Transport Spectral Unit		Automatic Headspace Injection Unit	
Analysis time	30 min	Inlet volume	100 μL
Column type	MXT-WAX, length 30 m, inside diameter 0.53 mm, film thickness 1 μm	Incubation time	10 min
Column temperature	60 °C	Incubation temperature	60 °C
Carrier Gas/Drift Gas	N_2_ (99.999% pure)	Injection needle temperature	85 °C
IMS temperature	45 °C	Incubation speed	500 r/min

**Table 3 foods-12-03345-t003:** The gradient profile of gas chromatography conditions.

Time	E1 (Drift Gas)	E2 (Carrier Gas)	Recording
00:00,000	150 mL/min	2 mL/min	rec
02:00,000		2 mL/min	-
10:00,000		10 mL/min	-
20:00,000		100 mL/min	-
30:00,000		100 mL/min	Stop

**Table 4 foods-12-03345-t004:** The basic physicochemical indicators of ten *A. arguta* wine samples.

Variety	Alcohol by Volume (*v*/*v*)	Residual Sugar (g/L)	Titratable Acid (g/L)	Dry Extract (g/L)	Vitamin C (mg/L)	Total Phenols (mg/L)	Total Flavonoids (mg/L)	Tannin (g/L)
‘Kuilv’	8.80 ± 0.20 ^bc^	1.49 ± 0.05 ^bcd^	13.07 ± 0.12 ^b^	48.01 ± 0.17 ^b^	913.46 ± 11.65 ^a^	816.10 ± 2.87 ^a^	304.14 ± 11.11 ^d^	1.35 ± 0.12 ^b^
‘Fenglv’	10.07 ± 0.21 ^a^	1.32 ± 0.04 ^de^	11.16 ± 0.07 ^d^	40.65 ± 0.12 ^c^	826.68 ± 10.68 ^c^	562.45 ± 1.66 ^f^	418.04 ± 2.57 ^b^	1.29 ± 0.05 ^b^
‘Jialv’	7.17 ± 0.15 ^e^	1.45 ± 0.05 ^bcd^	10.36 ± 0.19 ^f^	30.02 ± 0.25 ^e^	706.23 ± 15.54 ^d^	693.58 ± 11.95 ^c^	477.12 ± 7.27 ^a^	1.73 ± 0.12 ^b^
‘Wanlv’	9.13 ± 0.25 ^b^	1.58 ± 0.09 ^bc^	10.77 ± 0.11 ^e^	31.62 ± 0.17 ^d^	661.27 ± 1.96 ^e^	781.64 ± 2.87 ^b^	277.48 ± 8.40 ^e^	1.74 ± 0.16 ^b^
‘Xinlv’	8.93 ± 0.14 ^bc^	1.61 ± 0.21 ^b^	12.69 ± 0.13 ^c^	40.19 ± 0.17 ^c^	231.29 ± 3.48 ^i^	488.73 ± 15.99 ^g^	335.65 ± 4.20 ^c^	1.20 ± 0.23 ^b^
‘Pinglv’	9.83 ± 0.15 ^a^	1.37 ± 0.01 ^cde^	8.86 ± 0.20 ^g^	23.30 ± 0.29 ^f^	305.63 ± 2.06 ^h^	305.92 ± 6.63 ^i^	219.32 ± 4.20 ^f^	1.16 ± 0.12 ^b^
‘Lvbao’	8.10 ± 0.20 ^d^	1.92 ± 0.05 ^a^	10.29 ± 0.09 ^f^	12.25 ± 0.25 ^h^	432.46 ± 1.68 ^f^	797.92 ± 8.91 ^ab^	335.65 ± 4.87 ^c^	1.78 ± 0.28 ^b^
‘Cuiyu’	8.37 ± 0.15 ^cd^	1.21 ± 0.03 ^e^	10.85 ± 0.15 ^de^	23.09 ± 0.17 ^f^	394.65 ± 1.33 ^g^	375.79 ± 4.39 ^h^	202.36 ± 4.17 ^g^	4.63 ± 0.37 ^a^
‘Tianxinbao’	9.20 ± 0.26 ^b^	1.84 ± 0.12 ^a^	11.14 ± 0.17 ^d^	49.59 ± 0.30 ^a^	861.28 ± 1.11 ^b^	659.12 ± 5.98 ^d^	301.72 ± 12.59 ^d^	4.28 ± 0.42 ^a^
‘Longcheng No2’	8.93 ± 0.21 ^bc^	1.83 ± 0.03 ^a^	13.81 ± 0.14 ^a^	20.24 ± 0.30 ^g^	869.58 ± 2.52 ^b^	598.82 ± 4.39 ^e^	224.17 ± 4.20 ^f^	4.46 ± 0.04 ^a^

Means with different letters in the same column indicate significant differences (Duncan’s test, *p* < 0.05).

**Table 5 foods-12-03345-t005:** *CIELab* parameters of ten *A. arguta* wine samples.

Variety	*L**	*a**	*b**	*C***_ab_*	*h***_ab_*	Δ*E***_ab_*
‘Kuilv’	31.65 ± 0.02 ^a^	2.48 ± 0.01 ^e^	5.08 ± 0.03 ^a^	5.66 ± 0.04 ^a^	64.01 ± 0.09 ^a^	
‘Fenglv’	30.95 ± 0.02 ^e^	2.73 ± 0.02 ^b^	4.02 ± 0.01 ^f^	4.88 ± 0.02 ^d^	65.19 ± 0.33 ^a^	1.33 ± 0.02 ^d^
‘Jialv’	31.18 ± 0.06 ^d^	2.60 ± 0.05 ^d^	4.44 ± 0.10 ^d^	5.14 ± 0.11 ^c^	61.14 ± 0.80 ^a^	0.84 ± 0.11 ^e^
‘Wanlv’	31.58 ± 0.01 ^b^	2.34 ± 0.03 ^g^	5.06 ± 0.03 ^a^	5.58 ± 0.02 ^a^	64.16 ± 0.03 ^a^	0.19 ± 0.01 ^h^
‘Xinlv’	31.29 ± 0.02 ^c^	2.63 ± 0.01 ^cd^	4.59 ± 0.02 ^c^	5.29 ± 0.02 ^b^	60.19 ± 0.03 ^a^	0.66 ± 0.02 ^f^
‘Pinglv’	30.14 ± 0.01 ^f^	2.88 ± 0.01 ^a^	3.15 ± 0.00 ^i^	4.27 ± 0.01 ^f^	47.59 ± 0.03 ^c^	2.51 ± 0.00 ^a^
‘Lvbao’	30.98 ± 0.03 ^e^	2.82 ± 0.02 ^a^	3.75 ± 0.03 ^g^	4.70 ± 0.03 ^e^	53.03 ± 0.05 ^b^	1.56 ± 0.03 ^c^
‘Cuiyu’	31.59 ± 0.01 ^b^	2.38 ± 0.02 ^fg^	4.30 ± 0.01 ^e^	4.91 ± 0.01 ^d^	61.07 ± 0.17 ^a^	0.83 ± 0.01 ^e^
‘Tianxinbao’	31.22 ± 0.01 ^d^	2.67 ± 0.02 ^c^	3.41 ± 0.01 ^h^	4.33 ± 0.01 ^f^	51.97 ± 0.18 ^b^	1.76 ± 0.01 ^b^
‘Longcheng No2′	31.65 ± 0.03 ^a^	2.43 ± 0.03 ^ef^	4.81 ± 0.04 ^b^	5.38 ± 0.04 ^b^	63.34 ± 0.11 ^a^	0.31 ± 0.05 ^g^

Means with different letters in the same column indicate significant differences (Duncan’s test, *p* < 0.05).

**Table 6 foods-12-03345-t006:** Organic acids of ten *A. arguta* wine samples.

Variety	Oxalic Acid (g/L)	Quinic Acid (g/L)	Malic Acid (g/L)	Shikimic Acid (g/L)	Lactic Acid (g/L)	Acetic Acid (g/L)	Citric Acid (g/L)	Succinic Acid (g/L)
‘Kuilv’	0.05 ± 0 ^de^	4.2 ± 0.31 ^d^	0.83 ± 0.05 ^f^	0.05 ± 0 ^b^	0.22 ± 0.02 ^d^	0.63 ± 0.06 ^ef^	6.52 ± 0.19 ^b^	2.41 ± 0.24 ^a^
‘Fenglv’	0.07 ± 0 ^a^	4.27 ± 0.32 ^d^	2.02 ± 0.11 ^b^	N.A.	0.14 ± 0.01 ^g^	1.1 ± 0.12 ^a^	4.95 ± 0.33 ^e^	N.A.
‘Jialv’	0.04 ± 0 ^ef^	3.62 ± 0.17 ^ef^	1.77 ± 0.13 ^c^	0.01 ± 0 ^e^	0.13 ± 0.02 ^h^	0.76 ± 0.11 ^cd^	3.45 ± 0.32 ^f^	1.53 ± 0.21 ^b^
‘Wanlv’	0.01 ± 0 ^g^	3.54 ± 0.34 ^f^	1.07 ± 0.12 ^e^	0.06 ± 0 ^a^	0.24 ± 0.05 ^c^	0.53 ± 0.01 ^f^	5.39 ± 0.41 ^c^	0.32 ± 0.02 ^d^
‘Xinlv’	0.06 ± 0.01 ^bcd^	5.34 ± 0.46 ^b^	1.49 ± 0.23 ^d^	N.A.	0.28 ± 0.04 ^b^	0.87 ± 0.03 ^b^	5.13 ± 0.41 ^d^	N.A.
‘Pinglv’	0.04 ± 0 ^ef^	2.82 ± 0.22 ^g^	1.67 ± 0.14 ^c^	N.A.	0.19 ± 0.01 ^f^	0.66 ± 0.05 ^e^	3.12 ± 0.31 ^g^	0.72 ± 0.01 ^c^
‘Lvbao’	0.06 ± 0 ^bc^	3.7 ± 0.33 ^e^	1.36 ± 0.11 ^d^	0.04 ± 0 ^c^	0.27 ± 0.05 ^b^	0.71 ± 0.02 ^de^	5.2 ± 0.61 ^d^	N.A.
‘Cuiyu’	0.05 ± 0.01 ^de^	5.04 ± 0.27 ^c^	1.72 ± 0.1 ^c^	N.A.	0.2 ± 0.01 ^e^	0.84 ± 0.02 ^cd^	5.06 ± 0.72 ^de^	N.A.
‘Tianxinbao’	0.03 ± 0 ^f^	3.71 ± 0.35 ^e^	2.92 ± 0.34 ^a^	0.01 ± 0 ^e^	0.8 ± 0.07 ^a^	0.7 ± 0.01 ^de^	3.2 ± 0.21 ^g^	N.A.
‘Longcheng No2′	0.07 ± 0 ^a^	7.04 ± 0.78 ^a^	1.12 ± 0.16 ^e^	0.02 ± 0 ^d^	0.2 ± 0.01 ^e^	0.57 ± 0.01 ^f^	7.45 ± 0.81 ^a^	0.35 ± 0.02 ^d^

Means with different letters in the same column indicate significant differences (Duncan’s test, *p* < 0.05). N.A.: not available.

**Table 7 foods-12-03345-t007:** Composition of volatile compounds in ten *A. arguta* wine samples.

No.	CAS#	Aromatic Substances	Volatile Compound Content of *A. arguta* Wine (μg/L)									
			‘Kuilv’	‘Fenglv’	‘Jialv’	‘Wanlv’	‘Xinlv’	‘Pinglv’	‘Lvbao’	‘Cuiyu’	‘Tianxinbao’	‘Longcheng No2’
1	C111273	1-Hexanol D	145.09 ± 2.46 ^c^	147.11 ± 2.01 ^c^	133.64 ± 3.87 ^d^	127.3 ± 2.07 ^d^	130.14 ± 0.68 ^d^	149.72 ± 3.87 ^c^	132.7 ± 3.19 ^d^	191.94 ± 3.56 ^a^	130.47 ± 1.83 ^d^	171.09 ± 4.63 ^b^
2	C111273	1-Hexanol M	9.29 ± 1.03 ^b^	9.91 ± 0.45 ^b^	8.5 ± 0.80 ^b^	8.89 ± 0.76 ^b^	9.66 ± 0.63 ^b^	8.91 ± 0.61 ^b^	8.93 ± 0.29 ^b^	9.95 ± 0.45 ^b^	18.77 ± 1.01 ^a^	9.28 ± 0.62 ^b^
3	C3391864	1-Octen-3-ol	55.09 ± 3.71 ^g^	123.52 ± 1.30 ^e^	133.86 ± 5.96 ^cd^	155.97 ± 0.20 ^a^	146.71 ± 1.47 ^b^	133.79 ± 1.90 ^cd^	136.58 ± 1.83 ^c^	102.73 ± 1.52 ^f^	110.3 ± 1.42 ^f^	126.58 ± 1.64 ^de^
4	C71410	1-Pentanol	303.24 ± 2.43 ^c^	318.58 ± 0.78 ^b^	274.69 ± 2.59 ^e^	259.09 ± 4.63 ^f^	268.42 ± 1.05 ^e^	300.9 ± 2.24 ^cd^	295.52 ± 2.68 ^d^	333.92 ± 2.73 ^a^	268.56 ± 1.34 ^e^	293.75 ± 1.85 ^d^
5	C78831	2-Methyl-1-propanol	2623.85 ± 22.25 ^d^	3183.39 ± 12.10 ^a^	2563.98 ± 26.25 ^e^	2578.19 ± 20.02 ^de^	2749.31 ± 24.17 ^c^	2819.84 ± 14.94 ^b^	2830.67 ± 42.55 ^b^	2852.69 ± 47.72 ^b^	2554.27 ± 42.17 ^e^	2767.72 ± 15.11 ^c^
6	C71238	Propanol	16.63 ± 1.00 ^e^	21.43 ± 0.94 c^d^	19.34 ± 0.27 ^de^	21.6 ± 2.25 ^cd^	18.83 ± 0.90 ^de^	28.15 ± 0.91 ^b^	19.66 ± 0.95 ^de^	23.31 ± 0.34 c	39.11 ± 5.06 ^a^	28.35 ± 1.19 ^b^
7	C123513	3-Methyl-1-butanol D	5649.02 ± 41.99 ^c^	6117.13 ± 7.42 ^a^	5452.48 ± 24.93 ^d^	4851.22 ± 3.23 ^f^	5449.16 ± 40.45 ^d^	5851.85 ± 31.64 ^b^	5603.09 ± 20.09 ^c^	5848.77 ± 13.68 ^b^	5286.4 ± 33.96 ^e^	5853.18 ± 29.09 ^b^
8	C123513	3-Methyl-1-butanol M	192.2 ± 3.29 ^bc^	205.6 ± 6.01 ^a^	181.63 ± 1.48 ^d^	165.9 ± 1.64 ^e^	190.93 ± 3.53 ^c^	199.08 ± 2.49 ^b^	196.63 ± 6.69 ^bc^	193.26 ± 1.42 ^bc^	164.74 ± 3.64 ^e^	183.62 ± 4.02 ^d^
9	C626891	4-Methyl-1-pentanol	6344.69 ± 32.84 ^f^	6872.77 ± 31.17 ^a^	6307.65 ± 26.88 ^f^	5909.16 ± 57.48 ^h^	6075.88 ± 9.42 ^g^	6546.95 ± 21.67 ^d^	6351.96 ± 12.58 ^f^	6747.97 ± 13.06 ^b^	6425.46 ± 19.77 ^e^	6675.33 ± 20.01 ^c^
10	C64175	Ethanol D	8151.3 ± 30.97 ^e^	8054.36 ± 37.78 ^f^	8037.14 ± 23.47 ^f^	7865.83 ± 34.79 ^g^	8450.76 ± 39.19 ^b^	8542.87 ± 30.77 ^a^	8422.18 ± 37.46 ^bc^	8345.27 ± 30.99 ^d^	8360.43 ± 9.23 ^d^	8369.77 ± 26.68 ^cd^
11	C64175	Ethanol M	1048.36 ± 21.15 ^de^	947.6 ± 6.40 ^f^	1081.72 ± 27.19 ^cd^	1058.4 ± 20.07 ^de^	934.56 ± 32.34 ^f^	1030.47 ± 24.05 ^e^	1126.24 ± 20.23 ^bc^	1242.37 ± 23.92 ^a^	1141.39 ± 35.61 ^b^	1155.72 ± 37.15 ^b^
12	C543497	Heptan-2-ol	44.67 ± 2.31 ^b^	52 ±1.72 ^a^	37.85 ±0.31 ^d^	37.33 ± 1.76 ^d^	35.64 ± 1.20 ^d^	43.87 ± 2.30 ^bc^	38.51 ± 2.86 ^d^	38.83 ± 3.58	40.02 ± 1.29 ^cd^	36.85 ± 3.76 ^d^
Alcohols	Subtotal		24,583.41	26,053.39	24,232.49	23,038.88	24,459.99	25,656.39	25,162.66	25,931.00	24,539.92	25,671.24
	Percentage		58.50%	59.23%	57.59%	57.18%	57.59%	59.14%	59.35%	60.09%	58.64%	60.98%
1	C108645	Ethyl isovalerate	1198.58 ± 16.51 ^de^	1349.46 ± 9.09 ^a^	1145.87 ± 16.17 ^f^	1180.59 ± 6.31 ^de^	1211.71 ± 25.04 ^cd^	1259.43 ± 31.36 ^b^	1237.8 ± 25.04 ^bc^	1240.61 ± 31.36 ^bc^	1172.35 ± 20.84 ^ef^	1241.42 ± 18.72 ^bc^
2	C123864	Butyl acetate	61.56 ±3.98 ^e^	61.14 ±2.92 ^e^	61.41 ±3.09 ^e^	107.41 ± 3.01 ^b^	72.67 ±4.06 ^d^	75.15 ±1.42 ^d^	60.72 ± 2.37 ^e^	191.15 ± 0.60 ^a^	103.37 ± 2.72 ^b^	91.61 ± 1.01 ^c^
3	C109217	Butyl butanoate	430.68 ± 5.20 ^g^	586.35 ±5.15 ^a^	486.49 ± 6.20 ^f^	508.48 ± 5.14 ^c^	498.34 ± 3.59 ^d^	499.25 ± 6.93 ^d^	512.5 ± 3.70 ^c^	510.66 ± 0.59 ^c^	514.09 ± 5.29 ^c^	534.23 ± 3.94 ^b^
4	C591684	Butyl pentanoate	1077.42 ± 7.44 ^a^	1056.86 ± 7.04 ^ab^	1028.18 ± 11.02 ^b^	1026.08 ± 19.38 ^b^	1037.14 ± 32.04 ^b^	1030.17 ± 6.87 ^b^	962.33 ± 38.21 ^c^	968.18 ± 7.32 ^c^	949.46 ± 7.34 ^c^	847.15 ± 2.48 ^d^
5	C106365	Propyl propanoate	849.69 ± 8.57 ^b^	855 ± 4.21 ^ab^	862.66 ± 2.36 ^a^	725.97 ± 3.67 ^e^	753.5 ± 1.50 ^d^	851.73 ± 7.96 ^ab^	688.08 ± 11.31 ^f^	731.76 ± 7.8 ^e^	647.75 ± 6.29 ^g^	804.99 ± 3.53 ^c^
6	C141786	Ethyl acetate	3724.14 ± 11.93 ^b^	3527.12 ± 24.55 ^e^	3764.08 ± 24.44 ^a^	3552.94 ± 32.66 ^de^	3575.24 ± 21.14 ^d^	3684.5 ± 13.56 ^c^	3538.78 ± 6.77 ^e^	3582.56 ± 10.85 ^d^	3445.42 ± 10.96 ^f^	3693.5 ± 5.31 ^bc^
7	C105544	Ethyl butanoate	945.79 ± 1.95 ^c^	704.05 ± 11.05 ^f^	992.39 ± 2.44 ^b^	518.57 ± 1.50 ^h^	915.19 ± 5.89 ^d^	673.36 ± 3.51 ^g^	817.04 ± 4.19 ^e^	1253.93 ± 39.15 ^a^	714.35 ± 2.77 ^f^	816.73 ± 2.59 ^e^
8	C623701	Ethyl crotonate	292.46 ± 4.19 ^e^	346.73 ± 3.37 ^b^	285.51 ± 5.96 ^f^	273.79 ± 0.84 ^g^	363.13 ± 1.64 ^a^	309.71 ± 5.30 ^d^	243.5 ± 5.11 ^h^	273.34 ± 5.14 ^g^	319.28 ± 5.13 ^c^	233.9 ± 1.82 ^i^
9	C109944	Ethyl formate	28.50 ± 1.85 ^g^	73.86 ± 3.36 ^b^	40.2 ± 3.12 ^f^	58.6 ± 2.21 ^cd^	85.91 ± 6.08 ^a^	52.87 ± 3.01 ^e^	63.65 ± 0.05 ^c^	55.11 ± 4.01 ^de^	86.44 ± 3.30 ^a^	24.22 ± 0.24 ^g^
10	C106309	Ethyl heptanoate	82.52 ± 6.25 ^g^	91.64 ± 6.46 ^f^	96.75 ± 4.79 ^ef^	107.41 ± 5.53 ^d^	142.18 ± 4.58 ^b^	101.89 ± 5.93 ^de^	103.74 ± 4.95 ^de^	143.7 ± 5.41 ^b^	215.64 ± 2.30 ^a^	123.69 ± 5.99 ^c^
11	C123660	Ethyl hexanoate	1193.43 ± 38.82 ^b^	944.86 ± 31.15 ^d^	1236.77 ± 18.10 ^a^	1143.27 ± 34.24 ^c^	1135.09 ± 15.98 ^c^	849.78 ± 16.17 ^ef^	1106.02 ± 7.84 ^c^	564.1 ± 5.50 ^g^	865.11 ± 12.96 ^e^	823.85 ± 2.15 ^f^
12	C97621	Ethyl isobutyrate	10.94 ± 0.68 ^d^	20.02 ± 0.86 ^b^	15.31 ± 1.04 ^c^	12.4 ± 0.52 ^cd^	12.55 ± 0.29 ^cd^	20.78 ± 4.37 ^b^	15.85 ± 0.27 ^c^	14.32 ± 1.07 ^cd^	22.66 ± 0.11 ^b^	30.81 ± 3.87 ^a^
13	C97643	Ethyl lactate	26.94 ± 2.46 ^b^	32.61 ± 2.09 ^a^	32.65 ± 2.31 ^a^	35.73 ± 2.45 ^a^	35.13 ± 1.88 ^a^	34.32 ± 1.48 ^a^	35.98 ± 1.20 ^a^	35.52 ± 1.35 ^a^	27.57 ± 0.94 ^b^	33.15 ± 2.46 ^a^
14	C539822	Ethyl pentanoate	175.07 ± 4.43 ^d^	204.57 ± 4.98 ^c^	170.41 ± 0.79 ^d^	204.16 ± 3.76 ^c^	214.25 ± 8.36 ^b^	177.8 ± 2.03 ^d^	217.95 ± 5.67 ^b^	195.69 ± 3.03 ^c^	235.02 ± 8.11 ^a^	179.98 ± 7.89 ^d^
15	C105373	Ethyl propanoate	156.7 ± 2.90 ^g^	205.86 ± 0.48 ^e^	226.22 ± 3.87 ^d^	307.81 ± 1.34 ^c^	189.52 ± 4.48 ^f^	205.99 ± 2.85 ^e^	466.85 ± 1.69 ^a^	194.97 ± 5.97 ^f^	329.05 ± 1.80 ^b^	138.19 ± 3.99 ^h^
16	C142927	Hexyl acetate	123.26 ± 2.82 ^bc^	117 ± 1.70 ^cd^	130.5 ± 7.31 ^ab^	130.52 ± 2.94 ^ab^	126.17 ± 2.19 ^ab^	133.44 ± 3.12 ^a^	107.96 ± 2.78 ^e^	106.45 ± 3.04 ^e^	108.36 ± 7.52 ^e^	110.44 ± 6.37 ^de^
17	C123922	Isoamyl acetate	2557.4 ± 49.64 ^b^	2639.16 ± 31.28 ^a^	2437.83 ± 26.12 ^cd^	2239.54 ± 17.14 ^e^	2421.28 ± 31.75 ^d^	2478.3 ± 9.91 ^c^	2131.07 ± 14.08 ^g^	2096.37 ± 10.39 ^g^	2192.69 ± 5.34 ^f^	1864.17 ± 10.89 ^h^
18	C110190	Isobutyl acetate	523.86 ± 3.13 ^d^	741.09 ± 20.25 ^a^	482.42 ± 2.10 ^e^	445.05 ± 1.36 ^f^	556.1 ± 7.76 ^c^	601.03 ± 8.64 ^b^	484.68 ± 2.27 ^e^	551.64 ± 3.01 ^c^	481.35 ± 1.35 ^e^	359.86 ± 5.39 ^g^
19	C539902	Isobutyl butyrate	59.4 ± 1.55 ^ef^	86.59 ± 2.35 ^b^	64.14 ± 1.33 ^d^	58.2 ± 1.11 ^f^	64.47 ± 1.85 ^d^	76.33 ± 0.65 ^c^	62.43 ± 0.56 ^de^	64.73 ± 3.27 ^d^	76.7 ± 2.48 ^c^	101.90 ± 1.88 ^a^
20	C105680	Isopentyl propanoate	423.87 ± 6.91 ^cd^	483.67 ± 2.61 ^a^	415.18 ± 2.34 ^d^	391.74 ± 4.40 ^e^	426.74 ± 2.86 ^c^	447.95 ± 8.17 ^b^	401.61 ± 10.32 ^e^	418.44 ± 5.63 ^cd^	394.07 ± 2.51 ^e^	378.18 ± 2.86 ^f^
21	C868575	Methyl 2-methylbutanoate	213.31 ± 1.96 ^e^	223.06 ± 5.62 ^d^	226.27 ± 3.40 ^d^	270.46 ± 1.75 ^a^	261.1 ± 5.44 ^b^	243.13 ± 3.68 ^c^	273.05 ± 6.93 ^a^	212.89 ± 2.24 ^e^	262.03 ± 5.74 b	192.4 ± 6.02 ^f^
22	C79209	methyl acetate	748.92 ± 7.20 ^b^	648.33 ± 5.70 ^f^	730.81 ± 14.57 ^c^	709.44 ± 2.79 ^d^	751.35 ± 7.85 ^b^	754.46 ± 2.51 ^ab^	660.29 ± 8.40 ^e^	664.46 ± 3.20 ^e^	590.06 ± 1.40 ^g^	764.05 ± 2.26 ^a^
23	C106321	Ethyl octanoate D	16.08 ± 2.73 ^c^	19.93 ± 3.27 ^bc^	22.07 ±2.49 ^ab^	27.14 ± 0.89 ^a^	25.83 ± 1.89 ^b^	21.55 ± 2.27 ^abc^	22.48 ± 1.19 ^ab^	19.4 ± 4.59 ^bc^	19.09 ± 4.37 ^bc^	23.45 ± 3.62 ^ab^
24	C106321	Ethyl octanoate M	60.93 ± 7.20 ^g^	108.78 ± 8.37 ^f^	120.73 ± 2.89 ^e^	162.97 ± 3.13 ^a^	153.13 ± 3.51 ^abc^	152.4 ± 8.72 ^abc^	148.83 ± 4.59 ^bc^	141.75 ± 13.83 ^cd^	131.69 ± 2.79 ^de^	155.49 ± 3.87 ^ab^
Esters	Subtotal		14,981.45	15,127.74	15,074.84	14,198.27	15,027.73	14,735.32	14,363.20	14,231.72	13,903.61	13,566.64
	Percentage		35.65%	34.39%	35.83%	35.24%	35.38%	33.97%	33.88%	32.98%	33.22%	32.22%
1	C110623	Pentanal	37.91 ± 2.05 ^g^	53.26 ± 1.71 ^f^	66.77 ± 2.67 ^e^	101.84 ± 2.68 ^b^	88.51 ± 3.77 ^c^	65.58 ± 1.66 ^e^	79.42 ± 2.50 ^d^	83.6 ± 4.47 ^d^	134.93 ± 2.95 ^a^	32.88 ± 1.98 ^h^
2	C111717	Heptanal	13.44 ± 1.06 ^ef^	13.94 ± 0.47 ^ef^	12.26 ± 1.60 ^f^	16.51 ± 1.23 ^de^	32.56 ± 4.04 ^b^	15.37 ± 1.54 d^ef^	14.39 ± 1.76 ^ef^	17.84 ± 0.61 ^cd^	37.79 ± 2.23 ^a^	20.38 ± 1.06 ^c^
3	C66251	Hexanal	192.18 ± 2.31 ^bc^	186.89 ± 1.80 ^c^	198.24 ± 1.84 ^ab^	196.24 ± 3.27 ^ab^	195.51 ± 2.01 ^ab^	194.37 ± 4.13 ^b^	195.05 ± 3.35 ^b^	187.43 ± 5.35 ^c^	198.82 ± 5.66 ^ab^	202.06 ± 5.62 ^a^
4	C124130	Octanal	85.93 ± 12.73 ^f^	117.02 ± 5.30 ^de^	93.48 ± 4.96 ^f^	121.07 ± 1.94 ^cde^	117.56 ± 2.62 ^de^	128.36 ± 2.68 ^bc^	111.44 ± 3.73 ^e^	125.21 ± 1.65 b^cd^	156.49 ± 2.92 ^a^	133.7 ± 4.35 ^b^
5	C142836	(E,E)-2,4-hexadienal	40.72 ± 4.94 ^e^	108.47 ± 2.20 ^b^	110 ± 5.47 ^b^	129.49 ± 0.94 ^a^	132.42 ± 2.28 ^a^	127.06 ± 1.54 ^a^	102.34 ± 3.23 ^bc^	98.88 ± 2.22 ^cd^	94.92 ± 1.61 ^cd^	89.97 ± 2.87 ^d^
6	C105577	Diethyl acetal D	320 ± 13.28 ^c^	275.63 ± 10.33 ^e^	364.41 ± 6.50 ^a^	338.29 ± 7.61 ^b^	294.38 ± 3.72 ^d^	302.45 ± 7.07 ^d^	336.75 ± 5.71 ^b^	327.97 ± 3.40 ^bc^	271.06 ± 1.80 ^e^	364.93 ± 1.29 ^a^
7	C105577	Diethyl acetal M	298.7 ± 11.03 ^a^	171.82 ± 3.05 ^fg^	264.18 ± 8.81 ^b^	219.36 ± 4.15 ^d^	180.93 ± 0.61 ^f^	163.6 ± 4.34 ^g^	193.85 ± 3.40 ^e^	196.96 ± 3.16 ^e^	172.08 ± 4.30 ^fg^	248.68 ± 3.77 ^c^
8	C96173	2-Methylbutanal	173.69 ± 2.21 ^e^	194.93 ± 3.14 ^c^	164 ± 1.70 ^f^	196.06 ± 2.47 ^c^	197.36 ± 8.01 ^c^	163.5 ± 0.25 ^f^	145.2 ± 4.28 ^g^	221.99 ± 1.95 ^b^	277.63 ± 2.97 ^a^	180.68 ± 1.48 ^d^
9	C123728	Butanal	821.14 ± 20.72 ^e^	1137.16 ± 51.77 ^b^	964.03 ± 10.12 ^d^	1060.05 ± 12.62 ^c^	1149.04 ± 27.91 ^ab^	1196 ± 10.10 ^a^	1132.99 ± 19.49 ^b^	1084.15 ± 11.53 ^c^	1161.91 ± 50.29 ^ab^	1002.75 ± 12.98 ^d^
Aldehydes	Subtotal		1983.7	2259.11	2237.36	2378.93	2388.27	2356.28	2311.42	2344.04	2505.63	2276.03
	Percentage		4.72%	5.14%	5.32%	5.90%	5.62%	5.43%	5.45%	5.43%	5.99%	5.41%
1	C64197	Acetic acid D	153.84 ± 3.00 ^a^	126.51 ± 5.79 ^c^	126.04 ± 3.78 ^c^	129.06 ± 5.21 ^c^	117.59 ± 0.89 ^d^	138.17 ± 2.54 ^b^	152.28 ± 1.00 ^a^	151.22 ± 0.96 ^a^	153.64 ± 2.77 ^a^	142.22 ± 1.07 ^b^
	Subtotal		153.84	126.51	126.04	129.06	117.59	138.17	152.28	151.22	153.64	142.22
Acids	Percentage		0.37%	0.29%	0.30%	0.32%	0.28%	0.32%	0.36%	0.35%	0.37%	0.34%
1	C18402830	(E)-3-Nonen-2-one	12.36 ± 1.19 ^cd^	21.21 ± 0.98 ^a^	12.71 ± 0.83 ^cd^	12.43 ± 0.45 ^cd^	14.69 ± 0.69 ^c^	18.02 ± 1.18 ^b^	12.88 ± 0.24 ^cd^	10.38 ± 0.74 ^d^	13.33 ± 1.24 ^c^	17.58 ± 0.44 ^b^
2	C107879	2-Pentanone	46.88 ± 2.35 ^g^	82.75 ± 1.40 ^d^	72.22 ± 2.76 ^e^	125.4 ± 7.16 ^b^	96.18 ± 3.50 ^c^	76.96 ± 3.16 ^e^	61.12 ± 2.49 ^f^	128.05 ± 1.46 ^b^	293.08 ± 0.21 ^a^	86.21 ± 1.53 ^d^
3	C108101	4-Methyl-2-pentanone	117.19 ± 2.25 ^c^	134.00 ± 3.42 ^a^	125.48 ± 1.61 ^b^	104.6 ± 0.40 ^d^	118.5 ± 2.44 ^c^	132.94 ± 3.94 ^a^	96.48 ± 3.19 ^e^	106.17 ± 4.65 ^d^	92.44 ± 6.03 ^e^	96.41 ± 2.52 ^e^
Ketones	Subtotal		176.43	237.96	210.41	242.43	229.37	227.92	170.48	244.61	398.86	200.19
	Percentage		0.42%	0.54%	0.50%	0.60%	0.54%	0.53%	0.40%	0.57%	0.95%	0.48%
1	C100425	Styrene	30.03 ± 1.41 ^ab^	31.08 ± 0.32 a	29.08 ± 2.42 ^abc^	24.12 ± 1.26 ^cd^	29.52 ± 1.93 ^ab^	28.5 ± 1.89 ^abc^	24.92 ± 4.34 ^bcd^	29.21 ± 4.87 ^abc^	22.55 ± 2.40 ^d^	27.42 ± 2.72 ^abcd^
2	C586629	Terpinolene	112.65 ± 5.18 ^g^	151.71 ± 3.31 ^f^	167.81 ± 5.18 ^e^	276.89 ± 6.71 ^b^	223.02 ± 14.22 ^d^	239.74 ± 9.20 ^c^	212.51 ± 8.23 ^d^	218.97 ± 4.8 ^d^	326.22 ± 7.55 ^a^	216.72 ± 1.55 ^d^
Terpenoids	Subtotal		142.68	182.79	196.89	301.02	252.55	268.24	237.42	248.18	348.77	244.14
	Percentage		0.34%	0.42%	0.47%	0.75%	0.59%	0.62%	0.56%	0.58%	0.83%	0.58%
Total			42,021.49	43,987.49	42,078.03	40,288.58	42,475.51	43,382.33	42,397.46	43,150.77	41,850.43	42,100.46

Means with different letters in the same column indicate significant differences (Duncan’s test, *p* < 0.05). M and D refer to the monomers and dimers of the same substance.

**Table 8 foods-12-03345-t008:** OAV analysis of the main aroma compounds of ten *A. arguta* wine samples.

No.	Compound	Aroma Descriptors ^a^	‘Kuilv’	‘Fenglv’	‘Jialv’	‘Wanlv’	‘Xinlv’	‘Pinglv’	‘Lvbao’	‘Cuiyu’	‘Tianxinbao’	‘Longcheng No.2’
(Alcohols) 1	1-Octen-3-ol	Cucumber, Earth, Fat, Floral, Mushroom	2.75	6.18	6.69	7.80	7.34	6.69	6.83	5.14	5.52	6.33
2	4-Methyl-1-pentanol	Raw green flavor, Nut	1.55	1.68	1.54	1.44	1.48	1.6	1.55	1.65	1.57	1.63
(Esters) 1	Ethyl isovalerate	Apple, Fruit, Pineapple, Sour	39.95	44.98	38.2	39.35	40.39	41.98	41.26	41.35	39.08	41.38
2	Butyl butanoate	Floral	4.31	5.86	4.86	5.08	4.98	4.99	5.13	5.11	5.14	5.34
3	Ethyl butanoate	Apple, Butter, Cheese, Pineapple, Strawberry	47.29	35.2	49.62	25.93	45.76	33.67	40.85	62.7	35.72	40.84
4	Ethyl crotonate	Tropical Fruit	584.91	693.46	571.02	547.59	726.27	619.42	487.00	546.69	638.56	466.37
5	Ethyl hexanoate	Apple Peel, Brandy, Fruit Gum, Overripe Fruit, Pineapple	85.24	67.49	88.34	81.66	81.08	60.7	79	40.29	61.79	58.85
6	Ethyl isobutyrate	Apple, Fruit	<1	1.33	1.02	<1	<1	1.39	1.06	<1	1.51	2.05
7	Ethyl pentanoate	Apple, Dry Fish, Herb, Nut, Yeast	6.54	7.64	6.36	7.62	8	6.64	8.14	7.31	8.78	6.72
8	Isoamyl acetate	Apple, Banana, Glue, Pear	85.25	87.97	81.26	74.65	80.71	82.61	71.04	69.88	73.09	62.14
9	Isobutyl acetate	Apple, Banana, Floral, Herb	7.18	10.15	6.61	6.10	7.62	8.23	6.64	7.56	6.59	4.93
10	Ethyl octanoate D	Apricot, Brandy, Fat, Floral, Pineapple	3.22	3.99	4.41	5.43	5.17	4.31	4.5	3.88	3.82	4.69
11	Ethyl octanoate M	Apricot, Brandy, Fat, Floral, Pineapple	12.19	21.76	24.15	32.59	30.63	30.48	29.77	28.35	26.34	31.1
(Aldehydes) 1	2-Methylbutanal	Almond, Cocoa, Fermented, Hazelnut, Malt	133.61	149.94	126.15	150.82	151.81	125.77	111.69	170.76	213.57	138.99
2	Butanal	Banana, Green, Pungent	29.33	40.61	34.43	37.86	41.04	42.71	40.46	38.72	41.5	35.81
3	Octanal	Citrus, Fat, Green, Oil, Pungent	5.73	7.8	6.23	8.07	7.84	8.56	7.43	8.35	10.43	8.91
4	Pentanal	Almond, Bitter, Malt, Oil, Pungent	<1	<1	<1	<1	<1	<1	<1	<1	1.23	<1
(Terpenoids) 1	Terpinolene	Pine	2.75	3.7	4.09	6.75	5.44	5.85	5.18	5.34	7.96	5.29

Note: ^a^ From Flavornet database (https://www.femaflavor.org; http://www.flavornet.org; accessed on 12 July 2023).

## Data Availability

All related data and methods are presented in this paper. Additional inquiries should be addressed to the corresponding author.

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
