# Peer review of "Comprehensive Evaluation of Ten Actinidia arguta Wines Based on Color, Organic Acids, Volatile Compounds, and Quantitative Descriptive Analysis"

_foods, 2023, doi:10.3390/foods12183345_

Round 1

Reviewer 1 Report

The work is good but may be improved by the inclusion of the following suggestions. 

-          English should be improved throughout the manuscript.

-          Quantitative information should be provided in the abstract.

-          The concussion should be concise and to the point indicating the application of the work.

-          The novelty of the work should be established.

-     Please write one paragraph in the introduction about HPLC importance, in general, and you can consult the following articles to make this manuscript more useful to the readers.

J. Pham Biomed Anal 48, 175-188 (2001).; J. Sepn. Sci., 37, 1033-1057 (2014); J. Sepn. Sci., 35, 3235-3249 (2012). 

-          Please improve the quality of the Figures; especially 5 to onwards.

-          Please compare your results with previous studies and mention clearly how your work is important in comparison to already been reported.

no

Author Response

Thank you for your valuable and thoughtful comments.

Point 1: I English should be improved throughout the manuscript.

Response 1: Thank you for your valuable and thoughtful comments. We have carefully checked and improved the English writing in the revised manuscript.

Point 2: Quantitative information should be provided in the abstract.

Response 2: Quantitative information has been provided in the abstract. The modifications have been marked in red.

Point 3: The conclusion should be concise and to the point indicating the application of the work.

Response 3: We have simplified the conclusions and indicated the application of the work.

Point 4: The novelty of the work should be established.

Response 4: We have established the novelty of the work in the introduction.

Point 5: Please write one paragraph in the introduction about HPLC importance, in general, and you can consult the following articles to make this manuscript more useful to the readers.

Response 5: We have complemented the HPLC importance in the introduction to make this manuscript more useful to the readers.

Point 6: Please improve the quality of the Figures; especially 5 to onwards.

Response 6: We have replaced with higher resolution picture, and showed figure 8 and figure 9 in the Supplementary File at the end of the manuscript.

Point 7: Please compare your results with previous studies and mention clearly how your work is important in comparison to already been reported.

Response 7: We have revised it.

Reviewer 2 Report

The article is very informative and interesting. The authors performed detailed analyses for 10 varieties of A. arguta wine. The paper is very well written and organized. However many comments should be revised.

1-     In the abstract,

Full names for all abbreviations should be stated when first mentioning OAV,QDA

What is the suggestions for improving the quality of  A. arguta wine ?? the answer should be stated clearly in the abstract.

The novelty of the article should be highlighted over old similar studies

2-     The following articles should be cited in the introduction

Bioactivity, phytochemical profile and pro-healthy properties of Actinidia arguta: A review

https://doi.org/10.1016/j.foodres.2020.109449

Free and bound volatile compounds in ‘Hayward’ and ‘Hort16A’ kiwifruit and their wines

https://link.springer.com/article/10.1007/s00217-020-03452-9

Evaluation of the quality of fermented kiwi wines made from different kiwifruit cultivars

https://doi.org/10.1016/j.fbio.2021.101051

3-     Full name for GC-IMS, in line 68 should be stated. Also , abbreviations in line 88; VIP

4-     Lines 76-77, while there 76 are fewer studies examining the differences of A. arguta varieties on the aroma of A. arguta 77 wine.

These methods should be stated in more details with proper references.

5-     In line 81, In this study, 10 A. arguta varieties; the geographical origin should be stated

6-     Figure for the plant should be provided , may be in supplementary file

7-     Items 2.3.1- 2.3.4 need appropriate references

8-     Numbering of item 2.3.4 should be corrected to be 2.3.3 and consequently all proceeding items

9-     In line 219, the authors should state why they select 4-methyl-2-pentanol as the internal standard ?

10-  Titles for tables 2 and 3 should be stated in more details to describe the content exactly

11-  Resolution of figures 3, 4, and 5, should be improved or it can be displayed with original size in supplementary file

12-   2 tables are named as table 7. Correct numbering and stated the title in more details. Besides, some tables can be moved for supplementary file to decrease paper length.

13-  In line 444, correct table number; As can be seen from Table 6, also in line 452.

14-  Furthermore, the future research plan and limitations should be added at the end of the discussion.

15-  Conclusion is very informative but it should be written with full names for all abbreviations

16-  In funding, name of country should be provided.

Best wishes 

Author Response

Thank you for your valuable and thoughtful comments.

Point 1: In the abstract,

Full names for all abbreviations should be stated when first mentioning OAV,QDA.

What is the suggestions for improving the quality of  A. arguta wine ?? the answer should be stated clearly in the abstract.

The novelty of the article should be highlighted over old similar studies

Response 1: Thank you for your valuable and thoughtful comments. We have shown the full names of OAV and QDA.

We have added the suggestions for improving the quality of  A. arguta wine  in the abstract.

We have established the novelty of the work in the introduction.

Point 2: The following articles should be cited in the introduction

Bioactivity, phytochemical profile and pro-healthy properties of Actinidia arguta: A review

https://doi.org/10.1016/j.foodres.2020.109449

Free and bound volatile compounds in ‘Hayward’ and ‘Hort16A’ kiwifruit and their wines

https://link.springer.com/article/10.1007/s00217-020-03452-9

Evaluation of the quality of fermented kiwi wines made from different kiwifruit cultivars

https://doi.org/10.1016/j.fbio.2021.101051

Response 2: We have cited these articles in the introduction.

Point 3: Full name for GC-IMS, in line 68 should be stated. Also , abbreviations in line 88; VIP

Response 3: We have stated the full names of GC-IMS in line 68,and VIP in line 88.

Point 4: Lines 76-77, while there 76 are fewer studies examining the differences of A. arguta varieties on the aroma of A. arguta 77 wine. These methods should be stated in more details with proper references.

Response 4: We have restated these methods and added references.

Point 5: In line 81, In this study, 10 A. arguta varieties; the geographical origin should be stated.

Response 5: We have stated  the geographical origin of 10 A. arguta varieties.

Point 6: Figure for the plant should be provided , may be in supplementary file

Response 6: Figure for the plant have been provided in supplementary file at the end of the manuscript.

Point 7: Items 2.3.1- 2.3.4 need appropriate references

Response 7: In items 2.3.1 - 2.3.4 we have added appropriate references.

Point 8: Numbering of item 2.3.4 should be corrected to be 2.3.3 and consequently all proceeding items

Response 8: We have corrected it.

Point 9: In line 219, the authors should state why they select 4-methyl-2-pentanol as the internal standard?

Response 9: The reasons for selecting 4-methyl-2-pentanol as the internal standard are stated in item 2.3.5.

Point 10: Titles for tables 2 and 3 should be stated in more details to describe the content exactly

Response 10: We have described the titles of Tables 2 and 3 in more detail.

Point 11: Resolution of figures 3, 4, and 5, should be improved or it can be displayed with original size in supplementary file

Response 11: Figures 3, 4, and 5 have displayed with original size in supplementary file.

Point 12: 2 tables are named as table 7. Correct numbering and stated the title in more details. Besides, some tables can be moved for supplementary file to decrease paper length.

Response 12: We have corrected it.

Point 13: In line 444, correct table number; As can be seen from Table 6, also in line 452.

Response 13: We have corrected it.

Point 14: Furthermore, the future research plan and limitations should be added at the end of the discussion.

Response 14: We have added the future research plan and limitations at the end of the discussion.

Point 15: Conclusion is very informative but it should be written with full names for all abbreviations

Response 15: We have added the full names or all abbreviations.

Point 16: In funding, name of country should be provided.

Response 16: We have provided name of country in funding.

Round 2

Reviewer 1 Report

Revision is not complete; especially the novelty, language and necessary literature citation.

Either reject or give one more chance.

Revision is not complete; especially the novelty, language and necessary literature citation.

Either reject or give one more chance.

Author Response

Thank you for your careful checks. We are sorry for our carelessness. Based on the reviewers' and academic editors' comments, we tried our best to improve the manuscript and made some changes. These changes will not affect the content and framework of the paper. Here, we have emphasized these changes in red font in the revised manuscript. We would like to express our heartfelt thanks to all of you for your enthusiastic work and hope that the revised manuscript will be approved by you.

Point 1: Language

Response 1: We revised the English of the manuscript using the editing services listed at https://www.mdpi.com/authors/english. The changes were indicated in purple font.

Point 2: Novelty

Response 2:

In introduction, we have revised the manuscript novelty.

Although a series of works on A. arguta wine has been carried out by other researchers—such as on the effects of saccharomyces cerevisiae, process optimization, and varieties on the quality of A. arguta wine—studies on the flavor of A. arguta wine need improvement. First, HS-GC-IMS has been widely used to determine volatile compounds in food. However, the use of HS-GC-IMS to determine the volatile substances in A. arguta wines has not been reported. Second, previous research has only analyzed the volatile compounds in A. arguta wine and failed to identify the key volatile compounds. Last, there has been a lack of sensory evaluation, which is how customers can best evaluate the quality of A. arguta wine. In this study, the basic physicochemical indexes, color indexes and volatile compounds of A. arguta wine were determined and sensory evaluation was performed. The quality of different varieties of A. arguta wines was comprehensively evaluated. The fingerprints of volatile compounds in A. arguta wines were established by HS-GC-IMS. Moreover, based on the volatile compounds via multivariate statistical analysis quantitative descriptive analysis data, the specific A. arguta wine aroma characteristics were characterized while combined with principal component analysis, OAV value analysis, and VIP value analysis to screen the key volatile compounds affecting A. arguta wine aroma and identify the volatile compounds that may affect wine flavor.

Point 3: necessary literature citation

Response 3: We have endeavored to add some necessary references.

We have also made the following major changes to the article based on comments from the academic editor:

1.Title: we revised the title of the manuscript to ‘Comprehensive evaluation of ten Actinidia arguta wines based on color, organic acids, volatile compounds and quantitative descriptive analysis’.

2.In section 3.5: We have tried our best to find literature to compare the results of this study with previous literature. And the different families of volatile compounds are also discussed. In the multivariate statistics section, a number of references have also been added and revised.

3.In 3.8 section:we have correlated the sensory analysis with the key aroma substances obtained from the previous screening. Statistical analysis has also indicating that the eight compounds with VIP values greater than one calculated based on the OAV values were compounds that could influence the differences between groups, similar to the results of the sensory evaluation.

4. Conclusions: We have reorganized and revised it in response to comments from academic editor.

Reviewer 2 Report

the authors did all required  changes, thanks. only one note to follow journal style in reference writing. 

best wishes  

Author Response

Thank you for your careful checks. We are sorry for our carelessness. Based on the reviewers' and academic editors' comments, we tried our best to improve the manuscript and made some changes. These changes will not affect the content and framework of the paper. Here, we have emphasized these changes in red font in the revised manuscript. And we revised the English of the manuscript using the editing services listed at https://www.mdpi.com/authors/english. The changes were indicated in purple font. We would like to express our heartfelt thanks to all of you for your enthusiastic work and hope that the revised manuscript will be approved by you.

Point 1: Only one note to follow journal style in reference writing.

Response 1: Thank you for your valuable suggestions! We modified the references according to the foods journal style.

We have also made the following major changes to the article based on comments from the academic editor:

1.Title: we revised the title of the manuscript to ‘Comprehensive evaluation of ten Actinidia arguta wines based on color, organic acids, volatile compounds and quantitative descriptive analysis’.

2.In section 3.5: We have tried our best to find literature to compare the results of this study with previous literature. And the different families of volatile compounds are also discussed. In the multivariate statistics section, a number of references have also been added and revised.

3.In 3.8 section:we have correlated the sensory analysis with the key aroma substances obtained from the previous screening. Statistical analysis has also indicating that the eight compounds with VIP values greater than one calculated based on the OAV values were compounds that could influence the differences between groups, similar to the results of the sensory evaluation.

4.Conclusions: We have reorganized and revised it in response to comments from academic editor.